# The quantification of Simpson's paradox and other contributions to contingency table theory

**Friedrich Teuscher**[ORCID] *

Institute of Genetics and Biometry, Research Institute for Farm Animal Biology (FBN), Dummerstorf, Germany

* teuscher@fbn-dummerstorf.de

## Abstract

The analysis of contingency tables is a powerful statistical tool used in experiments with categorical variables. This study improves parts of the theory underlying the use of contingency tables. Specifically, the linkage disequilibrium parameter as a measure of two-way interactions applied to three-way tables makes it possible to quantify Simpson's paradox by a simple formula. With tests on three-way interactions, there is only one that determines whether the partial interactions of all variables agree or whether there is at least one variable whose partial interactions disagree. To date, there has been no test available that determines whether the partial interactions of a certain variable agree or disagree, and the presented work closes this gap. This work reveals the relation of the multiplicative and the additive measure of a three-way interaction. Another contribution addresses the question of which cells in a contingency table are fixed when the first- and second-order marginal totals are given. The proposed procedure not only detects fixed zero counts but also fixed positive counts. This impacts the determination of the degrees of freedom. Furthermore, limitations of methods that simulate contingency tables with given pairwise associations are addressed.

## 1 Introduction

Categorical variables are observed in many branches of science. Contingency table theory serves to infer such data. A great spectrum of analytical methods was presented by Agresti [1]. In the present paper, some parts of the theory are improved and some methods are added.

In their historical overview, Fienberg and Rinaldo [2] recognized Bartlett's [3] important contribution to the theory of contingency tables. Simpson [4] clarified some remaining questions from Bartlett's'paper on the three-way interaction in a 2×2×2 table. In addition to theoretical results, Simpson gave an example in which the health benefits of a drug appeared separately in both males and females. However, if the data were merged, no effect was seen. Furthermore, Blyth [5] showed that the merged data might even indicate a strong negative effect of the drug. This phenomenon was called "Simpson's paradox".

Several examples have been found in real life, demonstrating the principle's great practical relevance and the many different situations in which it may arise. Many studies have

**Data Availability Statement:** All relevant data are within the paper.

**Funding:** The author received no specific funding for this work.

**Competing interests:** NO authors have competing interests.

investigated how to circumvent this paradox, how best to deal with it, or how to interpret it (e.g., [6–14]). However, no short and elucidating presentation has so far succeeded in showing the relation of the paradox and the inner structure of the table.

Different measures are used for the association between two categorical variables, particularly for a 2×2 table (odds ratio, Yule's Q, Pearson's $\varphi$ and $\rho$). Quantitative genetics, for example, uses the so-called linkage disequilibrium (LD). The application of LD to the two-way and partial associations delivers a closed formula quantifying Simpson's paradox. The formula is derived in Section 2 and applied in Section 7.1, and it allows a clear, correct, and straightforward interpretation of a famous Berkeley data set.

In a strong sense, Bartlett [3] did not investigate a "three"-way interaction but a "third"-way interaction for a 2×2×2 table. He considered the question of whether a third variable (sex) has an effect on the association between the other two variables (success and treatment). He suggested comparing the odds ratios of the partial 2×2 tables (one for males and one for females). When they agree, the third variable has no effect.

Simpson [4] realized that Bartlett's definition of no three- (or third-) way interaction implies a symmetry property: when the third variable has no effect on the interaction between variables one and two (agreeing odds ratios of both sub-tables), then automatically, the first variable has no effect on the interaction between variables two and three, and the second variable has no effect on the interaction between variables one and three. Therefore, Bartlett's [3] test on "no three-way interaction" is a global one, and the alternative hypothesis would be "there is at least one variable with three-way interaction". Although such a test is not senseless at all, it is hard to believe that someone is interested in whether the interaction between treatment and sex for the group of successful patients equals the interaction between treatment and sex for the group of failed patients.

Therefore, a test for a single variable ("sex has no influence on the effect of a drug" versus "the effect of the drug differs between males and females") is still needed. It is clear that, for such a test, the odds ratio is not a suitable measure. A measure of association is needed that does not have the symmetry property. Simpson [4] mentioned that symmetry is lost for the root mean square contingency parameter, what we now call the correlation coefficient. However, he did not investigate this measure. It appears that this important issue has not been treated elsewhere so far, possibly because it does not fit the hierarchical log-linear model approach. In Section 3, this gap in the theory is closed. The method is applied to the Berkeley data in Section 7.2.

In quantitative genetics, the concept of LD has been generalized to three and four variables as the so-called three- or four-locus LD [15–20]. The three-locus LD of Bennett [15] is an additive measure and related to the additive measure of Lancaster [21, 22]. It was shown [23, 24] that this measure is not consistent with Bartlett's criterion, which is actually the solution of a cubic equation.

Bartlett's criterion, although appearing intuitive, turned out to agree with the maximum-likelihood equation of the log-linear model. Streitberg [25, 26] discussed the shortcomings of the log-linear model, treated the tables as multinomial distributions, and argued for additive measures. Obviously (and unfortunately), he was not aware of the investigations into tables and entropy performed by Good [27].

Shannon's [28] principle of entropy is a successful concept in physics, engineering, information theory, and statistics. Khinchin [29] delivered mathematical foundations for this principle. In particular, he investigated a measure $H$ for the information content of an experiment (with a finite size $n$ of possible events) as a functional of the probability function. The higher the value of $H$, the lower the information content of the experiment. He made two assumptions: (i) $H$ is largest when the events have unique probabilities $1/n$ and (ii) if an experiment

consists of two experimental parts, A and B, then the information content of the whole experiment, $H(AB)$, should be the sum of the information content of the first part $H(A)$ and the information content of the second part, given the first part, denoted by $H(B|A)$, i.e.,

$$H(A\ B) = H(A) + H(B|A). \tag{1}$$

He showed that, under these reasonable assumptions, there is only one measure that is continuous: the entropy $H = -\lambda \sum_{i=1}^{n} p_i \ln p_i$, where $\lambda$ is a positive constant and often set to one, i.e.,

$$H = -\sum_{i=1}^{n} p_i \ln p_i. \tag{2}$$

Good [27] treated contingency tables as multinomial distributions and determined the distribution with maximum entropy and given restraints, such as one- and two-way marginals. It turned out that his solution for 2×2×2 tables agreed with Bartlett's criterion.

There is another point speaking against Bennett's linear measure. In genetic multi-locus linkage analyses, Hill [23] showed that a table with an absence of three-way interactions may have negative "probabilities". That is, given a table with a three-way interaction, the corresponding hypothetical table without a three-way interaction would not exist. Such a dilemma cannot arise by applying the entropy principle because of its concavity.

It can be concluded that, for 2×2×2 tables, the multiplicative measure has a deeper impact than the additive one. On the other hand, the additive measure is much more tractable. Therefore, we ask which additive measure comes nearest (is most similar) to the multiplicative one. Section 4 examines whether Bennett's measure is the first-order Taylor expansion of Bartlett's measure.

A central theme in the progress of contingency table theory is the introduction and development of the log-linear model. In their historical overview, Fienberg and Rinaldo [2] show that a special point was the difficulty in handling zero counts. The nonexistence of the maximum-likelihood estimator (MLE) was indicated by the lack of convergence of the algorithms used to compute the MLE. Later, Fienberg and Rinaldo [30, 31] generated a numerical procedure specifically designed to check for the existence of the MLE. They based their approach on investigations of extended exponential families and the geometrical properties of log-linear models. Practically, the question about zero counts was whether the marginal totals enforce the cells to have zero counts. In such cases, the cell is fixed and this therefore also influences the degrees of freedom. So far, it has been overlooked that not only zero count cells but also positive count cells might be fixed. Section 5 presents an elementary algorithm that detects all fixed cells.

There are variables with categories that have an obvious order, and such variables are called ordinally scaled. [32–35] documented the progress and problems with simulating ordinally scaled variables with given pairwise Pearson's correlation coefficients. The techniques are modifications and adaptations of simulation techniques for multivariate normally distributed variables with a given correlation matrix. However, there are no procedures available that work for every admissible correlation matrix. Section 6 presents a simulation method that has no such theoretical limitations.

[35] handled the same task but with demanded pairwise associations measured with Goodman and Kruskal's $\gamma$. [36] generated a program for Lee's procedure. Although the authors did not mention it, the method is not suitable for simulating all admissible scenarios. These shortcomings are overcome in Section 6.

In Section 7, a real data set reflecting Simpson's paradox is analyzed with tools derived in Sections 2 and 3.

The paper concludes with a discussion of the issues. Special attention is given to the application of the entropy principle.

## 2 The quantification of Simpson's paradox

Let $\underline{X}$ and $\underline{Y}$ be two random categorical variables with $I_X$ and $I_Y$ categories, respectively. In an experiment, $n$ objects are inspected to identify which categories of variables $\underline{X}$ and $\underline{Y}$ apply. The counts $n_{i,j}$, $i = 1,2,\cdots,I_X$, $j = 1,2,\cdots,I_Y$, are written in a $I_X \times I_Y$ contingency table. The probability that an object matches categories $X_i$ and $Y_j$ is $p_{i,j} = P(\underline{X} = X_i \wedge \underline{Y} = Y_j)$, and its estimate is $n_{i,j}/n$. The association between categories $X_i$ and $Y_j$ is defined by the linkage disequilibrium (LD) measures:

$$D_{X_i,Y_j} := p_{i,j} - p_{i,\bullet}p_{\bullet,j}, \quad i = 1, 2, \cdots, I_X, j = 1, 2, \cdots, I_Y. \tag{3}$$

The point indicates summation over the assigned variable, e.g., $p_{i,\bullet} = \sum_{j=1}^{I_Y} p_{i,j}$, delivering marginal probabilities.

The LD $D_{X_i,Y_j}$ is assigned to the pair $(X_i, Y_i)$ of categories. The relation of this pair to all other pairs can be summarized by collapsing the $I_X \times I_Y$ table into the 2×2 table $\begin{bmatrix} p_{i,j} & p_{i,\bar{j}} \\ p_{\bar{i},j} & p_{\bar{i},\bar{j}} \end{bmatrix}$, where the bar over an index means summation over all categories with exception of the category defined by the index. The 2×2 table then takes the form

$$\begin{bmatrix} p_{i,j} & p_{i,\bar{j}} \\ p_{\bar{i},j} & p_{\bar{i},\bar{j}} \end{bmatrix} = \begin{bmatrix} p_{i,j} & p_{i,\bullet} - p_{i,j} \\ p_{\bullet,j} - p_{i,j} & 1 - p_{i,\bullet} - p_{\bullet,j} + p_{i,j} \end{bmatrix}. \tag{4}$$

It is easy to check that $D_{X_i,Y_j} = p_{i,j} p_{\bar{i},\bar{j}} - p_{i,\bar{j}} p_{\bar{i},j}$ holds. Pearson's correlation coefficient can then be written as

$$\rho_{X_i,Y_j} = \frac{D_{X_i,Y_j}}{\sqrt{p_{i,\bullet}(1 - p_{i,\bullet})p_{\bullet,j}(1 - p_{\bullet,j})}}, \tag{5}$$

which coincides with Pearson's $\varphi$.

With $\underline{Z}$ being a third categorical variable, the cell probabilities of the associated $I_X \times I_Y \times I_Z$ table are now $p_{i,j,k} = P(\underline{X} = X_i \wedge \underline{Y} = Y_j \wedge \underline{Z} = Z_k)$, $k = 1,2,\cdots,I_Z$. Eq (3) then change to

$$D_{X_i,Y_j} = p_{i,j,\bullet} - p_{i,\bullet,\bullet}p_{\bullet,j,\bullet}. \tag{6}$$

Using the definition of conditional probabilities, $p_{i,j|k} = p_{i,j,k}/p_{\bullet,\bullet,k}$, the conditional analogue to Eq (3) is

$$\begin{aligned} D_{X_i,Y_j|Z_k} = p_{i,j|k} - p_{i,\bullet|k}p_{\bullet,j|k} &= \frac{p_{i,j,k}}{p_{\bullet,\bullet,k}} - \frac{p_{i,\bullet,k}p_{\bullet,j,k}}{p_{\bullet,\bullet,k}^2} = \\ &= \frac{p_{i,j,k}}{p_{\bullet,\bullet,k}} - \frac{(D_{X_i,Z_k} + p_{i,\bullet,\bullet}p_{\bullet,\bullet,k})(D_{Y_j,Z_k} + p_{\bullet,j,\bullet}p_{\bullet,\bullet,k})}{p_{\bullet,\bullet,k}^2} = \\ &= \frac{p_{i,j,k}}{p_{\bullet,\bullet,k}} - p_{i,\bullet,\bullet}p_{\bullet,j,\bullet} - \frac{D_{X_i,Z_k}D_{Y_j,Z_k}}{p_{\bullet,\bullet,k}^2} - \frac{p_{\bullet,j,\bullet}D_{X_i,Z_k+p_{i,\bullet,\bullet}D_{Y_j,Z_k}}}{p_{\bullet,\bullet,k}} \end{aligned} \tag{7}$$

Because $\sum_{k=1}^{I_Z} D_{X_i,Z_k} = \sum_{k=1}^{I_Z}(p_{i,\bullet,k} - p_{i,\bullet,\bullet}p_{\bullet,\bullet,k}) = p_{i,\bullet,\bullet} - p_{i,\bullet,\bullet}p_{\bullet,\bullet,\bullet} = 0$ and, analogously, $\sum_{k=1}^{I_Z} D_{Y_j,Z_k} = 0$, the weighted sum $\bar{D}_{X_i,Y_j|Z}$ is

$$\bar{D}_{X_i,Y_j|Z} = \sum_{k=1}^{I_Z} p_{\bullet,\bullet,k} D_{X_i,Y_j|Z_k} =$$

$$= \sum_{k=1}^{I_Z} p_{\bullet,\bullet,k}\left(\frac{p_{i,j,k}}{p_{\bullet,\bullet,k}} - p_{i,\bullet,\bullet}p_{\bullet,j,\bullet} - \frac{D_{X_i,Z_k}D_{Y_j,Z_k}}{p_{\bullet,\bullet,k}^2} - \frac{p_{\bullet,j,\bullet}D_{X_i,Z_k} + p_{i,\bullet,\bullet}D_{Y_j,Z_k}}{p_{\bullet,\bullet,k}}\right) =$$

$$= \sum_{k=1}^{I_Z}\left(p_{i,j,k} - p_{\bullet,\bullet,k}p_{i,\bullet,\bullet}p_{\bullet,j,\bullet} - \frac{D_{X_i,Z_k}D_{Y_j,Z_k}}{p_{\bullet,\bullet,k}} - \frac{p_{\bullet,j,\bullet}D_{X_i,Z_k} + p_{i,\bullet,\bullet}D_{Y_j,Z_k}}{1}\right) =$$

$$= p_{i,j,\bullet} - p_{i,\bullet,\bullet}p_{\bullet,j,\bullet} - p_{\bullet,j,\bullet}\sum_{k=1}^{I_Z} D_{X_i,Z_k} - p_{i,\bullet,\bullet}\sum_{k=1}^{I_Z} D_{Y_j,Z_k} - \sum_{k=1}^{I_Z}\frac{D_{X_i,Z_k}D_{Y_j,Z_k}}{p_{\bullet,\bullet,k}} =$$

$$= D_{X_i,Y_j} - \sum_{k=1}^{I_Z}\frac{D_{X_i,Z_k}D_{Y_j,Z_k}}{p_{\bullet,\bullet,k}}.$$

The result is formulated as a theorem.

THEOREM: For an $I_X \times I_Y \times I_Z$ table, the difference between the two-way LD and the weighted sum of the partial LDs is

$$D_{X_i,Y_j} - \bar{D}_{X_i,Y_j|Z} = \sum_{k=1}^{I_Z}\frac{D_{X_i,Z_k}D_{Y_j,Z_k}}{p_{\bullet,\bullet,k}}, i \in \{1, 2, \cdots, I_X\}, j \in \{1, 2, \cdots, I_Y\}. \tag{8}$$

For a 2×2×2 table, the difference becomes

$$D_{X_i,Y_j} - \bar{D}_{X_i,Y_j|Z} = \frac{D_{X_i,Z_k}D_{Y_j,Z_k}}{p_{\bullet,\bullet,k}(1 - p_{\bullet,\bullet,k})}, \qquad i, j, k \in \{1, 2\}. \tag{9}$$

The simplification for the 2×2×2 table follows from inserting $I_Z = 2$ into Eq (8) and regarding the well-known formula $D_{A,B} = -D_{A,\bar{B}}$.

## 3 Testing the equality of partial interactions for one variable

The null hypothesis for Bartlett's test concerning 2×2×2 tables is the agreement of all partial interactions (measured as the odds ratio), while the alternative hypothesis is that at least one pair of partial interactions is unequal. Here, the effect (if any) of the third variable on the interaction between the first and the second variables is inferred. Let the first variable be the outcome of the experiment with categories "success" and "no success", the second variable be the applied treatment with categories "1" and "2" (one treatment could be a placebo), and the third variable be the sex of the patient with categories "male" and "female".

The null hypothesis is that both partial interactions of the third variable coincide. The hypothetical table with agreeing partial interactions and the observed table have several parameters in common: three two-way marginal totals, three one-way marginal totals, and the sample size

(zero-way marginal total). The equational system for the probabilities is then

$$
\begin{bmatrix} 1 \\ p_{1,\bullet,\bullet} \\ p_{\bullet,1,\bullet} \\ p_{\bullet,\bullet,1} \\ p_{1,1,\bullet} \\ p_{1,\bullet,1} \\ p_{\bullet,1,1} \end{bmatrix}
=
\begin{bmatrix} 1 \\ p_1 \\ p_2 \\ p_3 \\ p_{1,2} \\ p_{1,3} \\ p_{2,3} \end{bmatrix}
=
\begin{bmatrix}
1 & 1 & 1 & 1 & 1 & 1 & 1 & 1 \\
1 & 1 & 1 & 1 & 0 & 0 & 0 & 0 \\
1 & 1 & 0 & 0 & 1 & 1 & 0 & 0 \\
1 & 0 & 1 & 0 & 1 & 0 & 1 & 0 \\
1 & 1 & 0 & 0 & 0 & 0 & 0 & 0 \\
1 & 0 & 1 & 0 & 0 & 0 & 0 & 0 \\
1 & 0 & 0 & 0 & 1 & 0 & 0 & 0
\end{bmatrix}
\begin{bmatrix} p_{1,1,1} \\ p_{1,1,2} \\ p_{1,2,1} \\ p_{1,2,2} \\ p_{2,1,1} \\ p_{2,1,2} \\ p_{2,2,1} \\ p_{2,2,2} \end{bmatrix},
\tag{10}
$$

where the second vector defines the abbreviations of the first one. Solving system (10) gives

$$
\mathbf{p}(p_{1,1,1}) =
\begin{bmatrix} p_{1,1,1} \\ p_{1,1,2} \\ p_{1,2,1} \\ p_{1,2,2} \\ p_{2,1,1} \\ p_{2,1,2} \\ p_{2,2,1} \\ p_{2,2,2} \end{bmatrix}
=
\begin{bmatrix} p_{1,1,1} \\ -p_{1,1,1} \\ -p_{1,1,1} \\ p_{1,1,1} \\ -p_{1,1,1} \\ p_{1,1,1} \\ p_{1,1,1} \\ -p_{1,1,1} \end{bmatrix}
+
\begin{bmatrix} 0 \\ p_{1,2} \\ p_{1,3} \\ p_1 - p_{1,2} - p_{1,3} \\ p_{2,3} \\ p_2 - p_{1,2} - p_{2,3} \\ p_3 - p_{1,3} - p_{2,3} \\ 1 - p_1 - p_2 - p_3 + p_{1,2} + p_{1,3} + p_{2,3} \end{bmatrix}.
\tag{11}
$$

With eight cells and seven conditions, there is one free parameter, $p_{1,1,1}$. The partial tables for male and female patients are presented in Table 1.

There is certainly no effect due to sex if both sub-tables agree. Solving this system of linear equations gives $p_{1,2} = p_1 p_2$, $p_{1,3} = p_1 p_3$ and $p_{2,3} = p_2 p_3$; i.e., all variables were pairwise independent.

If the odds ratios of both sub-tables agree, this would apply also for the sub-tables of the other variables, as acknowledged by Simpson [4]. Hence, it would not be a specific property of sex.

Thinking about LD, which is a relative measure (since the maximum and minimum depend on the one-way marginals), and the correlation coefficient, the better measure for associations in agreement will be the correlation coefficient.

**Table 1. Success × treatment sub-tables for the sexes with given one- and two-way marginals.**

| | | Treatment 1 | Treatment 2 |
|---|---|---|---|
| Male | Success | $\frac{p_{1,1,1}}{p_3}$ | $\frac{p_{1,3} - p_{1,1,1}}{p_3}$ |
| | No Success | $\frac{p_{2,3} - p_{1,1,1}}{p_3}$ | $\frac{p_{1,1,1} - p_{1,3} - p_{2,3} + p_3}{p_3}$ |
| Female | Success | $\frac{p_{1,2} - p_{1,1,1}}{1 - p_3}$ | $\frac{p_{1,1,1} - p_{1,2} - p_{1,3} + p_1}{1 - p_3}$ |
| | No Success | $\frac{p_{1,1,1} - p_{1,2} - p_{2,3} + p_2}{1 - p_3}$ | $1 - \frac{p_{1,2} + p_{1,3} + p_{2,3} - p_1 - p_2 - p_{1,1,1}}{1 - p_3}$ |

Determination for both sub-tables gives

$$\rho_{1_1,2_1|3_1} = \frac{p_{1,1,1}p_3 - p_{1,3}p_{2,3}}{A} \text{ and} \tag{12}$$

$$\rho_{1_1,2_1|3_2} = \frac{(1-p_3)(p_{1,2} - p_{1,1,1}) - (p_1 - p_{1,3})(p_2 - p_{2,3})}{B} \tag{13}$$

with $A = \sqrt{p_{1,\bullet,1}p_{\bullet,1,1}p_{\bullet,2,1}p_{2,\bullet,1}}$ and $B = \sqrt{p_{1,\bullet,2}p_{\bullet,1,2}p_{\bullet,2,2}p_{2,\bullet,2}}$.

Solving $\rho_{1_1,2_1|3_1} = \rho_{1_1,2_1|3_2}$ with respect to $p_{1,1,1}$ finds

$$\tilde{p}_{1,1,1} = \frac{B\,p_{1,3}p_{2,3} + A\{(1-p_3)p_{1,2} - (p_1 - p_{1,3})(p_2 - p_{2,3})\}}{B\,p_3 \quad + \quad A(1-p_3)} = \frac{B\,p_{1,\bullet,1}p_{\bullet,1,1} + A(p_{\bullet,\bullet,2}p_{1,1,\bullet} - p_{1,\bullet,2}p_{\bullet,1,2})}{B\,p_{\bullet,\bullet,1} \quad + \quad A\,p_{\bullet,\bullet,2}}. \tag{14}$$

Now we have the observed table and, via Eq (11), the table $n\mathbf{p}(\tilde{p}_{1,1,1})$ under the null hypothesis (no sex effect). The $\chi^2$ test with one degree of freedom can be used for the decision between the null and alternative hypothesis (sex has an effect). A measure for the third-way interaction can be defined by setting $D_{1,2;3} = \tilde{p}_{1,1,1} - p_{1,1,1}$.

## 4 A linear expression for Bartlett's measure for three-way association

Bartlett's measure $D$ for a three-way association in a 2×2×2 table is determined by solving

$$(p_{1,1,1} - D)(p_{1,2,2} - D)(p_{2,1,2} - D)(p_{2,2,1} - D) = (p_{1,1,2} + D)(p_{1,2,1} + D)(p_{2,1,1} + D)(p_{2,2,2} + D) \tag{15}$$

for $D$. Here, the probabilities $p_{i,j,k}$ are assigned to counts $n_{i,j,k}$ by $p_{i,j,k} = n_{i,j,k}/n_{\bullet,\bullet,\bullet}$. Vanishing $D$ indicates the absence of a three-way association.

Bennett [15] introduced an additive measure of the three-way association:

$$L_{1,2,3} = p_{1,1,1} - (p_1\,p_2\,p_3 + p_1\,D_{2,3} + p_2\,D_{1,3} + p_3\,D_{1,2}). \tag{16}$$

In the introduction, it was concluded that the multiplicative measure (15) has more impact and the linear measure (16) could be an approximation. Therefore, it can be checked whether the linear measure could be the first-order Taylor expansion of the nonlinear one.

Substituting the LD expressions $p_{i,j} = D_{i,j} + p_i\,p_j$ for the two-way probabilities of Eq (11) leads to

$$\mathbf{p}(p_{1,1,1}) = \begin{bmatrix} p_{1,1,1} \\ p_{1,1,2} \\ p_{1,2,1} \\ p_{1,2,2} \\ p_{2,1,1} \\ p_{2,1,2} \\ p_{2,2,1} \\ p_{2,2,2} \end{bmatrix} = \begin{bmatrix} p_{1,1,1} \\ -p_{1,1,1} \\ -p_{1,1,1} \\ p_{1,1,1} \\ -p_{1,1,1} \\ p_{1,1,1} \\ p_{1,1,1} \\ -p_{1,1,1} \end{bmatrix} + \begin{bmatrix} 0 \\ D_{1,2} + p_1\,p_2 \\ D_{1,3} + p_1\,p_3 \\ p_1(1 - p_2 - p_3) - D_{1,2} - D_{1,3} \\ D_{2,3} + p_2\,p_3 \\ p_2(1 - p_1 - p_3) - D_{1,2} - D_{2,3} \\ p_3(1 - p_1 - p_2) - D_{1,3} - D_{2,3} \\ D_{1,2} + D_{1,3} + D_{2,3} + (1 - p_1)(1 - p_2 - p_3) + p_2\,p_3 \end{bmatrix}. \tag{17}$$

Inserting the cell probabilities into Eq (15) gives a cubic equation of argument $p_{1,1,1}$. Using Mathematica, the roots of (15) were determined and the first-order Taylor expansions at LDs

of zero and one-way probabilities of one half were carried out. The real solution was

$$
\begin{aligned}
p_{1,1,1}^{\text{(Taylor)}} = {} & 64(1-2p_1)(1-2p_2)(1-2p_3)D_{1,2}D_{1,3}D_{2,3} + 4(1-2p_1)D_{1,2}D_{1,3} + \\
& + 4(1-2p_2)D_{1,2}D_{2,3} + 4(1-2p_3)D_{1,3}D_{2,3} + p_3 D_{1,2} + p_2 D_{1,3} + p_1 D_{2,3} + p_1 p_2 p_3.
\end{aligned}
\tag{18}
$$

The appropriate measure for the three-way interaction would be $D^{\text{(Taylor)}} = p_{1,1,1} - p_{1,1,1}^{\text{(Taylor)}}$. It can be seen that Bennett's measure differs from this but covers the four simplest terms. Therefore, Bennett's criterion can be interpreted as a simplified version of the first-order Taylor expansion of Bartlett's criterion. The second-order expansion was also available. The only unexpected result was that one coefficient was not a power of 2: the largest term was $-29 \times 2^{12}(1-2p_1)(1-2p_2)(1-2p_3)D_{1,2}^2 \, D_{1,3}^2 \, D_{2,3}^2$. The other coefficients were the following (excluding linear terms): $2^{12}$ (three terms), $2^9$ (three terms), $2^8$ (three terms), and $2^5$ (six terms).

## 5 The determination of fixed cells

### 5.1 The application of linear programming

Assume an observed $I_1 \times I_2 \times \cdots \times I_c$ contingency table $\mathbf{n}$, i.e., there are $c$ categorical variables, with $I_i$ categories per variable $i$, $i \in (1, 2, \cdots, c)$. The contingency table $\mathbf{n}$ is characterized by the counts $n_{i_1, i_2, \cdots, i_c}$, with $n = n_{\bullet, \ldots, \bullet} = \sum_{i_1, i_2, \cdots, i_c} n_{i_1, i_2, \cdots, i_c}$ counts overall. The one-way marginal totals $n_{i_{k_i}} = n_{\bullet, \ldots, \bullet, k_i, \bullet, \ldots, \bullet}$ can be written as

$$
n_{i_{k_i}} = \sum_{k_1=1}^{I_1} \sum_{k_2=1}^{I_2} \cdots \sum_{k_{i-1}=1}^{I_{i-1}} \sum_{k_{i+1}=1}^{I_{i+1}} \cdots \sum_{k_c=1}^{I_c} n_{k_1, k_2, \ldots, k_{i-1}, k_i, k_{i+1}, \ldots, k_c},
\tag{19}
$$

where $k_i$, $1 \leq k_i \leq I_i$, is a category of variable $i$.

Analogously, the two-way marginal totals $n_{i_{k_i} j_{k_j}} = n_{\bullet, \ldots, \bullet, k_i, \bullet, \ldots, \bullet, k_j, \bullet, \ldots, \bullet}$ can be written as

$$
n_{i_{k_i} j_{k_j}} = \sum_{k_1=1}^{I_1} \cdots \sum_{k_{i-1}=1}^{I_{i-1}} \sum_{k_{i+1}=1}^{I_{i+1}} \cdots \sum_{k_{j-1}=1}^{I_{j-1}} \sum_{k_{j+1}=1}^{I_{j+1}} \cdots \sum_{k_c=1}^{I_c} n_{k_1, k_2, \cdots, k_{i-1}, k_i, k_{i+1}, \cdots, k_{j-1}, k_j, k_{j+1}, \cdots, k_c},
\tag{20}
$$

where $i$ and $j$ define the involved variables and $k_i$ and $k_j$ define the appropriate categories.

The marginal totals with indices $I_i$, $i \in \{1, 2, \cdots, c\}$ can be determined by the others:

$$
\begin{aligned}
n_{i_{I_i}} &= n - \sum_{k_i=1}^{I_i-1} n_{i_{k_i}}, \\
n_{i_{I_i} j_{I_j}} &= n_{i_{I_i}} - \sum_{k_j=1}^{I_j-1} n_{i_{I_i} j_{k_j}} = 1 - \sum_{k_i=1}^{I_i-1} n_{i_{k_i}} - \sum_{k_j=1}^{I_j-1} \Big( n_{j_{k_j}} - \sum_{k_i=1}^{I_i-1} n_{i_{k_i} j_{k_j}} \Big) = \\
&= 1 - \sum_{k_i=1}^{I_i-1} n_{i_{k_i}} - \sum_{k_j=1}^{I_j-1} n_{j_{k_j}} + \sum_{k_i=1}^{I_i-1} \sum_{k_j=1}^{I_j-1} n_{i_{k_i} j_{k_j}}.
\end{aligned}
\tag{21}
$$

Let $\mathbf{m}$ be the vector of the considered marginal totals. Then, $\mathbf{m}$ can be written as

$$
\mathbf{m} = (n, n_{1, \bullet, \ldots, \bullet}, \ldots, n_{I_1-1, \bullet, \ldots, \bullet}, n_{\bullet, \ldots, \bullet, 1}, \ldots, n_{\bullet, \ldots, \bullet, I_c-1}, n_{1,1, \bullet, \ldots, \bullet}, \ldots, n_{\bullet, \ldots, \bullet, I_{c-1}-1, I_c-1})'.
\tag{22}
$$

We order the cells of $\mathbf{n}$ into a vector $\overrightarrow{\mathbf{n}}$, with

$$
\begin{aligned}
\overrightarrow{\mathbf{n}} &= (\overrightarrow{n}_1, \overrightarrow{n}_2, \ldots, \overrightarrow{n}_d)' = \\
&= (n_{1,1,\ldots,1}, \ldots, n_{1,\ldots,1,I_c}, n_{1,\ldots,1,2,1}, \ldots, n_{1,\ldots,1,2,I_c}, \ldots, n_{1,\ldots,1,I_{c-1},I_c}, \ldots, n_{I_1,I_2,\ldots,I_{c-1},I_c})',
\end{aligned}
\tag{23}
$$

and thus establish a one-to-one relation between $\overrightarrow{n}_j$ and $n_{i_1, i_2, \ldots, i_{c-1}, i_c}$, with

$j = \sum_{m=1}^{c-1} \{(i_m - 1) \prod_{k=i+1}^{c} I_k\} + i_c$. If $j$ is given, the corresponding $c$-tuple $i_1, i_2, \ldots, i_{c-1}, i_c$ must be evaluated sequentially. The length of $\overrightarrow{\mathbf{n}}$ is $d = \prod_{k=1}^{c} I_k$.

Then, the restraints can be formulated as

$$\mathbf{m} = A \overrightarrow{\mathbf{n}}, \tag{24}$$

where matrix $A$ has entries zero or one and ensures the addition of the demanded components of $\overrightarrow{\mathbf{n}}$. For a 2×2×2 table, matrix $A$ can be seen in Eq (10). In the first row of $A$, there are only ones, ensuring the addition of all components of $\overrightarrow{\mathbf{n}}$. In the second row, there is a one if the corresponding component of $\overrightarrow{\mathbf{n}}$ has category one at the first variable, etc.

We now introduce a table $\tilde{\mathbf{n}}$ with the same dimensions as the observed table $\mathbf{n}$. As with Eq (23), the unknown cell counts $\tilde{n}_{i_1, i_2, \cdots, i_c}$ are ordered into a vector $\overleftarrow{\mathbf{n}}$, with

$$
\begin{aligned}
\overleftarrow{\mathbf{n}} &= (\overleftarrow{n}_1, \overleftarrow{n}_2, \ldots, \overleftarrow{n}_d)' = \\
&= (\tilde{n}_{1,1,\ldots,1}, \ldots, \tilde{n}_{1,\ldots,1,I_c}, \tilde{n}_{1,\ldots,1,2,1}, \ldots, \tilde{n}_{1,\ldots,1,2,I_c}, \ldots, \tilde{n}_{1,\ldots,1,I_{c-1},I_c}, \ldots, \tilde{n}_{I_1,I_2,\ldots,I_{c-1},I_c})'.
\end{aligned} \tag{25}
$$

Since we are looking for a table $\tilde{\mathbf{n}}$ that has the same zero-, one-, and two-way marginals as the observed table $\mathbf{n}$,

$$\mathbf{m} = A \overleftarrow{\mathbf{n}} \tag{26}$$

must be valid, where matrix $A$ is the same as before. Then, the set $S$ with

$$S = \{\tilde{\mathbf{n}} | \mathbf{m} = A \overleftarrow{\mathbf{n}} \wedge \overleftarrow{\mathbf{n}} \geq 0\}, \tag{27}$$

where $\overleftarrow{\mathbf{n}} \geq 0$ is meant componentwise and ensures nonnegativity, contains all admissible tables satisfying the constraints.

There is at least one element of $S$: the observed table $\mathbf{n}$. Assuming that there are two elements in $S$, $\mathbf{n}_1$ and $\mathbf{n}_2$, then the linear combination $\lambda \mathbf{n}_1 + (1-\lambda) \mathbf{n}_2$ with $0 \leq \lambda \leq 1$ is also admissible. Hence, the set of admissible tables is convex and, furthermore, the theory of linear optimization is applicable. In particular, there exist unique solutions (e.g., see [37]) for the linear optimization problems

$$\text{Maximize } \overleftarrow{n}_i \text{ under the conditions } \mathbf{m} = A \overleftarrow{\mathbf{n}} \text{ and } \overleftarrow{\mathbf{n}} \geq 0 \tag{28A}$$

and

$$\text{Minimize } \overleftarrow{n}_i \text{ under the conditions } \mathbf{m} = A \overleftarrow{\mathbf{n}} \text{ and } \overleftarrow{\mathbf{n}} \geq 0 \tag{28B}$$

with $i \in \{1, 2, \cdots, d\}$. An upper bound $b_i^u$ can be obtained from Eq (28A) and a lower bound $b_i^l$ from Eq (28B) for $\overleftarrow{n}_1 : b_i^l \leq \overleftarrow{n}_i \leq b_i^u$. This is of importance, since $b_i^l = b_i^u$ means that $\overleftarrow{n}_i$ is fixed. The aim is to find such components. Sequential checking of $b_i^l = b_i^u$, $i = 1, 2, \cdots, d$, leads to the set, say $\Omega$, of the components for which the equality is valid. Several numerical software packages contain a linear programming or optimization procedure. As an example, the 4×4×4 data from Table 6 of Fienberg and Ricardo [2] was analyzed, with the results presented in Table 2. There were 24 zero and 12 nonzero counts, which turned out to be fixed.

Table 2 provides more information (such as $\tilde{n}_{1,4,2} = 6 - \tilde{n}_{1,1,2}$) than can be obtained from the described algorithm. We will come to that now.

## 5.2 The application of algebraic software

Applying algebraic software such as Mathematica [38] to problems (28A) and (28B) has an advantage compared to pure numerical algorithms. The system of equations $\mathbf{m} = A \overleftarrow{\mathbf{n}}$ can be

**Table 2. Cell counts for given zero-, one-, and two-way marginal totals for Table 6 of Fienberg and Rinaldo [2].** (The original table is obtained by inserting $\tilde{n}_{1,1,2} = 4$ and $\tilde{n}_{1,1,3} = \tilde{n}_{2,3,1} = \tilde{n}_{3,2,1} = 1$).

| | | Cell counts $\tilde{n}_{i,j,k}$ | | | |
| --- | --- | --- | --- | --- | --- |
| | | $j = 1$ | $j = 2$ | $j = 3$ | $j = 4$ |
| $i = 1$ | $k = 1$ | 0 | 0 | 0 | 4 |
| | $k = 2$ | $\tilde{n}_{1,1,2}$ | 0 | 0 | $6 - \tilde{n}_{1,1,2}$ |
| | $k = 3$ | $\tilde{n}_{1,1,3}$ | $10 - \tilde{n}_{1,1,2} - \tilde{n}_{1,1,3}$ | 0 | $\tilde{n}_{1,1,2} - 2$ |
| | $k = 4$ | $6 - \tilde{n}_{1,1,2} - \tilde{n}_{1,1,3}$ | $\tilde{n}_{1,1,2} + \tilde{n}_{1,1,3}$ | 3 | 2 |
| $i = 2$ | $k = 1$ | 0 | 0 | $\tilde{n}_{2,3,1}$ | $3 - \tilde{n}_{2,3,1}$ |
| | $k = 2$ | $9 - \tilde{n}_{1,1,2}$ | 0 | $6 - \tilde{n}_{2,3,1}$ | $\tilde{n}_{1,1,2} + \tilde{n}_{2,3,1} - 3$ |
| | $k = 3$ | $1 + \tilde{n}_{1,1,2}$ | 3 | 4 | $6 - \tilde{n}_{1,1,2}$ |
| | $k = 4$ | 0 | 0 | 2 | 0 |
| $i = 3$ | $k = 1$ | 0 | $\tilde{n}_{3,2,1}$ | $4 - \tilde{n}_{2,3,1} - \tilde{n}_{3,2,1}$ | $2 + \tilde{n}_{2,3,1}$ |
| | $k = 2$ | 5 | 6 | $4 + \tilde{n}_{2,3,1}$ | $3 - \tilde{n}_{2,3,1}$ |
| | $k = 3$ | 0 | 2 | 0 | 0 |
| | $k = 4$ | 0 | $3 - \tilde{n}_{3,2,1}$ | $3 + \tilde{n}_{3,2,1}$ | 0 |
| $i = 4$ | $k = 1$ | 5 | $2 - \tilde{n}_{3,2,1}$ | $1 + \tilde{n}_{3,2,1}$ | 3 |
| | $k = 2$ | 1 | 0 | 0 | 0 |
| | $k = 3$ | $6 - \tilde{n}_{1,1,2} - \tilde{n}_{1,1,3}$ | $\tilde{n}_{1,1,2} + \tilde{n}_{1,1,3} - 3$ | 0 | 0 |
| | $k = 4$ | $\tilde{n}_{1,1,2} + \tilde{n}_{1,1,3} - 4$ | $6 - \tilde{n}_{1,1,2} - \tilde{n}_{1,1,3} - \tilde{n}_{3,2,1}$ | $4 - \tilde{n}_{3,2,1}$ | 0 |

solved using the procedure "Solve", thus decreasing the number of variables. For a 2×2×2 table, the solution of Eq (24) is

$$
\mathbf{x} = \mathbf{x}_h + \mathbf{x}_S = \begin{bmatrix} \tilde{n}_{1,1,1} \\ -\tilde{n}_{1,1,1} \\ -\tilde{n}_{1,1,1} \\ \tilde{n}_{1,1,1} \\ -\tilde{n}_{1,1,1} \\ \tilde{n}_{1,1,1} \\ \tilde{n}_{1,1,1} \\ -\tilde{n}_{1,1,1} \end{bmatrix} + \begin{bmatrix} 0 \\ n_{1,1,\bullet} \\ n_{1,\bullet,1} \\ n_{1,\bullet,\bullet} - n_{1,1,\bullet} - n_{1,\bullet,1} \\ n_{\bullet,1,1} \\ n_{\bullet,1,\bullet} - n_{1,1,\bullet} - n_{\bullet,1,1} \\ n_{\bullet,\bullet,1} - n_{1,\bullet,1} - n_{\bullet,1,1} \\ n - n_{1,\bullet,\bullet} - n_{\bullet,1,\bullet} - n_{\bullet,\bullet,1} + n_{1,1,\bullet} + n_{1,\bullet,1} + n_{\bullet,1,1} \end{bmatrix}. \tag{29}
$$

Here, the representation $\mathbf{x} = \mathbf{x}_h + \mathbf{x}_s$ is used [37], where $\mathbf{x}_h$ is the solution of the homogenous system, i.e., $\mathbf{0} = A\,\mathbf{x}_h$, and $\mathbf{x}_s$ is a special solution, i.e., $\mathbf{m} = A\,\mathbf{x}_s$. Eq (29) is just Eq (11) times $n$. From the eight initial variables of the table, there is only one left: $\tilde{n}_{1,1,1}$.

In the general case of an $I_1 \times I_2 \times \cdots \times I_c$ table, there are $d = \prod_{i=1}^{c} I_i$ cells or initial variables. In matrix $A$ of Eq (24), there are $\sum_{i=1}^{c}(I_i - 1)$ linear independent rows for the one-way totals, $\sum_{i=1}^{c-1}\{(I_i - 1)\sum_{j=i+1}^{c}(I_j - 1)\}$ linear independent rows for the two-way totals, and one row for the overall total $n$. The number, $r$, of linear independent rows is therefore

$$
r = 1 + \sum_{i=1}^{c}(I_i - 1) + \sum_{i=1}^{c-1}\{(I_i - 1)\sum_{j=i+1}^{c}(I_j - 1)\}, \text{ or} \tag{30}
$$

$$r = (c-2)\left(\frac{c-1}{2} - I_\bullet\right) + I_\bullet^2 - \sum_{i=1}^{c}\{I_i\sum_{j=1}^{i}I_j\}. \tag{31}$$

The system of linear equations $\mathbf{m} = A\overleftarrow{\mathbf{n}}$ can be solved for $r$ components of $\overleftarrow{\mathbf{n}}$. There remain $f = d-r$ free variables, i.e.,

$$f = \prod_{i=1}^{c}I_i - (c-2)\left(\frac{c-1}{2} - I_\bullet\right) - I_\bullet^2 + \sum_{i=1}^{c}\{I_i\sum_{j=1}^{i}I_j\}. \tag{32}$$

For a three-way table, investigated by Roy and Kastenbaum [39], this number is the well-known $f = (I_1-1)(I_2-1)(I_3-1)$. For $c\geq 3$ and a unique number of categories $I$, $f$ turns out to be

$$f = I^c - \frac{c(I-1)\{c(I-1) - I + 3\}}{2} - 1. \tag{33}$$

Let $\mathbf{y}$ be the vector of the $f$ free variables. Then, the linear optimization problems have the Mathematica forms

$$b_i^u = \text{Maximize}[\{\overleftarrow{n}_i, \overleftarrow{n}_1 \geq 0, \overleftarrow{n}_2 \geq 0, \ldots, \overleftarrow{n}_d \geq 0\}, \{y_1, y_2, \ldots, y_f\}] \tag{34A}$$

and

$$b_i^l = \text{Minimize}[\{\overleftarrow{n}_i, \overleftarrow{n}_1 \geq 0, \overleftarrow{n}_2 \geq 0, \ldots, \overleftarrow{n}_d \geq 0\}, \{y_1, y_2, \ldots, y_f\}] \tag{34B}$$

and must be calculated for $i = 1,\ldots,d$. Let $\Omega$ be the set $\Omega = \{i|b_i^l = b_i^u \wedge i \in \{1, \ldots, d\}\}$, i.e., the set of indices of fixed cells.

When $\Omega$ is not empty, the analysis can be refined. In Eq (26), the fixed counts substitute for the variables of the fixed cells. This means that vector $\overleftarrow{\mathbf{n}}$ in $\mathbf{m} = A\overleftarrow{\mathbf{n}}$ has to be renewed by setting $\overleftarrow{n}_i = b_i^l$, $i \in \Omega$. Analogously, the list of variables is reduced by canceling the variables of the fixed cells. The solution of the new system of equations then further reduces the number of free variables. Applying this to Table 6 of Fienberg and Rinaldo [2] yielded Table 2 of this study. Only four free variables are left over. In this way, a very compact expression for admissible tables was reached.

## 6 The simulation of ordinally scaled variables with predefined associations

### 6.1 Association measured by Pearson's $\rho$

The aim is to simulate an $I_1{\times}I_2{\times}\cdots{\times}I_c$ contingency table of $c$ numerical variables with given one-way marginals and Pearson's correlation coefficients $\rho_{i,j}$, $i,j{\in}\{1,2,\cdots,c\}$. The categories of the variables are characterized by numbers. These numbers, $v_i(i_{k_i})$, $i{\in}\{1,2,\cdots,c\}$, $k_i{\in}\{1,\ldots,I_i\}$, may agree with the index of the category, i.e., $v_i(i_1) = 1$, $v_i(i_2) = 2,\ldots$, $v_i(i_{I_i}) = I_i$ for variable $i$.

The contingency table is characterized by the probabilities $p_{k_1,k_2,\ldots,k_c}$, where $k_i{\in}\{1,\ldots,I_i\}$ defines the category of variable $i$. The probabilities are unknown at present. Now the equations are collected to ensure the validity of the given conditions.

The one-way probabilities, $p_{i_{k_i}}$, are assumed to be known. Here, $i$ is the number of the variable and $k_i$ is the category of that variable. As before, $p_{i_{k_i}}$ may be obtained from the $c$-way

probabilities, $p_{k_1,k_2,\dots,k_c}$, by summing over all indices except index $i$, i.e.,

$$p_{i_{k_i}} = p_{\bullet,\dots,\bullet,i_{k_i},\bullet,\dots,\bullet}. \tag{35}$$

Hence, the expectations $\mu_i = \sum_{k_i=1}^{I_i} v_i(k_i)p_{i_{k_i}}$ and the variances $\sigma_i^2 = \sum_{k_i=1}^{I_i} v_i(k_i)^2 p_{i_{k_i}} - \mu_i^2$ of the variables $i \in \{1,\dots,c\}$ are also known.

The pairwise correlation coefficients, $\rho_{i,j}$, are defined by

$$\rho_{i,j} = \frac{cov_{i,j}}{|\sigma_i||\sigma_j|} = \frac{\sum_{k_i=1}^{I_i} \sum_{k_j=1}^{I_j} [v_i(k_i) - \mu_i][v_j(k_j) - \mu_j]p_{i_{k_i}j_{k_j}}}{|\sigma_i||\sigma_j|} = \frac{\sum_{k_i=1}^{I_i} \sum_{k_j=1}^{I_j} v_i(k_i)v_j(k_j)p_{i_{k_i}j_{k_j}} - \mu_i\mu_j}{|\sigma_i||\sigma_j|}, \tag{36}$$

where $p_{i_{k_i}j_{k_j}}$ are the two-way probabilities of variables $i$ and $j$, which can be determined via $p_{i_{k_i}j_{k_j}} = p_{\bullet,\dots,\bullet,i_{k_i},\bullet,\dots,\bullet,j_{k_j},\bullet,\dots,\bullet}$. Hence,

$$\rho_{i,j}|\sigma_i||\sigma_j| + \mu_i\mu_j = \sum_{k_i=1}^{I_i}\sum_{k_j=1}^{I_j} v_i(k_i)v_j(k_j)p_{\bullet,\dots,\bullet,i_{k_i},\bullet,\dots,\bullet,j_{k_j},\bullet,\dots,\bullet} \tag{37}$$

must hold. The left-hand side of this equation only involves constants; i.e., the left-hand side is a constant. The right-hand side of Eq (37) is a linear combination of the cell probabilities; therefore, the theory of linear programming can be applied.

For convenience, Mathematica and the principles of Section 5.2 are used to proceed. First, the system $W$ of the linear equations ($1 = p_{\bullet,\dots,\bullet}$, $\sum_{i=1}^{c}(I_i - 1) = I_{\bullet} - c$ equations for the one-way marginals and $(c-1)c/2$ equations for the two-way correlations) is solved.

Then, the lower and upper bounds for the first free variable are determined using the procedures Minimize and Maximize of Mathematica. Three cases need to be considered. (1) The procedure finds no solution, in which case there is no table satisfying the demanded correlations (in the literature, there was no practical and sufficient criterion for the existence of a table). (2) The lower bound and upper bound agree. Then, the first free variable is fixed. (3) The lower and upper bound differ, so there are a variety of tables satisfying the demanded correlations. Therefore, it must be decided whether an average table or an extreme one is preferred. We suggest simulating at least an average table and possibly afterward simulating extreme tables. For the average table, we assign the mean of the bounds to the first free variable. For extreme tables, either the lower or the upper bound can be assigned to the first free variable.

In either case, the first free variable is now assigned to a constant value, and the system of equations $W$ is updated by inserting that value for the variable. Then, the lower and upper bounds for the new first free variable are determined. (The output "no solution" may no longer appear.) This algorithm is repeated until there are no free variables left and all cell probabilities are determined.

Now, we have all $d = \prod_{k=1}^{c} I_k$ cell probabilities. We interpret them to define a $d$–point distribution. Then, the inversion algorithm of Lee [35] can be used to simulate the table.

For the case with no solution for the restraints, admissible scenarios can be determined. Instead of maximizing and minimizing the cell probabilities, we determine the bounds for the correlation parameters. For example, let the one-way marginals of a 3×3×3 table be $\mathbf{p}_1 = (0.1, 0.3, 0.6)'$, $\mathbf{p}_2 = (0.2, 0.4, 0.4)'$, and $\mathbf{p}_3 = (0.3, 0.3, 0.4)'$. The appropriate expectations and variances are thereby defined. Then, the procedures Maximize and Minimize are used to calculate the bounds for the correlation parameters. The obtained bounds are $-0.797 \le \rho_{1,2} \le 0.797$, $-0.808 \le \rho_{1,3} \le 0.808$, and $-0.837 \le \rho_{2,3} \le 0.933$. Maximizing $\rho_{1,2}+\rho_{1,3}+\rho_{2,3}$ yields 2.537, with $\rho_{1,2} = 0.797$, $\rho_{1,3} = 0.808$, and $\rho_{2,3} = 0.933$. Minimizing $\rho_{1,2}+\rho_{1,3}+\rho_{2,3}$ yields $-1.400$, with

$\rho_{1,2} = -0.598$, $\rho_{1,3} = -0.449$, and $\rho_{2,3} = -0.354$. For $\rho = \rho_{1,2} = \rho_{1,3} = \rho_{2,3}$, we obtain the admissible interval $-0.598 \leq \rho \leq 0.797$.

## 6.2 Association measured by Goodman and Kruskal's $\gamma$

Lee [35] developed an algorithm for the simulation of a table with given one-way marginal totals and given pairwise association measures in terms of Goodman and Kruskal's $\gamma$. Ibrahim and Suliadi [36] provided a macro program of this algorithm.

This section is organized as follows. First, the algorithm of Lee [35] is described, including three improvements. Then, we use it in two examples showing scenarios of association parameters where a table satisfying the demands does not exist and, even when such a table exists, it cannot be determined with Lee's [35] method. Later, hints are provided for how to handle these problems.

Consider two ordinally scaled categorical variables $Y_1$ and $Y_2$ with $I_1$ and $I_2$ categories, respectively. Let the (unknown) joint probabilities be denoted by $p_{i,j} = P(Y_1 = i \wedge Y_2 = j)$. Consider two random objects with observations of both variables, $O_1 = (Y_1, Y_2)$ and $O_2 = (Y'_1, Y'_2)$. The probability that the first object has categories $i$ and $j$ and the second object has categories $i'$ and $j'$ is then $p_{i,j} p_{i',j'}$. In addition to being objects with observations, $O_1$ and $O_2$ are also two points of the $I_1 \times I_2$ table, and they may be concordant ($i < i'$ and $j < j'$ or $i > i'$ and $j > j'$), discordant ($i < i'$ and $j > j'$ or $i > i'$ and $j < j'$), or indifferent (at least one equality sign appears). Adding the concordant and the discordant cases, the definition of $\gamma$ becomes

$$S = \sum_{i<i'}\sum_{j<j'}p_{i,j}p_{i',j'} + \sum_{i>i'}\sum_{j>j'}p_{i,j}p_{i',j'},$$
$$D = \sum_{i<i'}\sum_{j>j'}p_{i,j}p_{i',j'} + \sum_{i>i'}\sum_{j<j'}p_{i,j}p_{i',j'}, \quad (38)$$
$$\gamma = \frac{S - D}{S + D}.$$

Note that this definition deviates from that of Lee [35] ones. (This is the first improvement by the author.) In the version from [35] or [40] the right-hand-side double sums do not appear. In that case, however, we can obtain differing association values if we rename or interchange the variables. Since this is not judicious in the actual context, the symmetrical version (38) is applied. However, this does not affect the ideas of Lee [35] in an essential way.

Following Lee [35], for given one-way marginals, i.e., for given $\mathbf{p} = (p_{1,\bullet}, p_{2,\bullet}, \ldots, p_{I_1,\bullet})'$ and $\mathbf{q} = (p_{\bullet,1}, p_{\bullet,2}, \ldots, p_{\bullet,I_2})'$, the maximum gamma is $\gamma = 1$. The probabilities $p_{i,j}$ carrying this property can be determined by the following routine. With an outer loop $i = 1,2,\cdots,I_1$ and for each $i$ with an inner loop $j = 1,2,\cdots,I_2$ (or vice versa), set

$$p_{i,j} = \min(\mathbf{p}_i, \mathbf{q}_j) \text{ and update } \mathbf{p}_i = \mathbf{p}_i - p_{i,j} \text{ and } \mathbf{q}_j = \mathbf{q}_j - p_{i,j}. \quad (39)$$

(This is the second improvement by the author. The author thought that Lee [35] meant the same, but his version was hard to understand.)

For negative gammas, the method must be modified. [35] and [36] stated that a two-way table with perfect negative association (i.e., $\gamma = -1$) can be obtained from the two-way table with perfect positive association (i.e., $\gamma = 1$) by reversing the order of categories for one of the variables. To see that this is not correct, consider a table with three variables where all association parameters are $\gamma = 1$. Reversing the order of categories of the first variable changes two associations to $\gamma = -1$. If one then reverses the order of categories of the second or third variable, there remain two associations with $\gamma = -1$ and one with $\gamma = 1$. If we then reverse the order of categories of the remaining variable that has not changed so far, we again have three associations of $\gamma = 1$. Therefore, a table with three variables, where all association parameters are $\gamma = -1$, cannot be generated.

However, the joint probabilities for $\gamma = -1$ can be determined by reversing the components of the one-way marginal totals $\mathbf{p}$, i.e., $\mathbf{p} = (p_{I_1,\bullet}, p_{I_1-1,\bullet}, \ldots, p_{1,\bullet})\prime$, applying routine (39), and next reversing the rows of matrix $\{p_{i,j}\}_{I_1 \times I_2}$, thus obtaining the table $\mathbf{p}^{\min}$ with the originally demanded one-way marginal totals. (This was the third improvement by the author.)

Denote the generated $I_1 \times I_2$ table with $\mathbf{p}^{\mathrm{opt}}$ and the $I_1 \times I_2$ table for independent $Y_1$ and $Y_2$ with $\mathbf{p}^0$, i.e., $p_{i,j}^0 = p_{i,\bullet} p_{\bullet,j}$. Then, the convex linear combination $\mathbf{p}(\lambda) = (1-\lambda)\mathbf{p}^0 + \lambda \mathbf{p}^{\mathrm{opt}}$, $0 \le \lambda \le 1$, defines a table $\mathbf{p}(\lambda)$ satisfying the one-way marginal totals. For $\lambda = 0$, $\mathbf{p}(0) = \mathbf{p}^0$ holds and the appropriate gamma is zero, i.e., $\gamma[\mathbf{p}(0)] = 0$. Also, for $\lambda = 1$, $\mathbf{p}(1) = \mathbf{p}^{\mathrm{opt}}$ holds and the appropriate gamma is one, i.e., $\gamma[\mathbf{p}(1)] = 1$. Since $\gamma[\mathbf{p}(\lambda)]$ is a continuous function of $\lambda$, there must be a $\lambda^*$ so that $\gamma[\mathbf{p}(\lambda^*)] = \Gamma$, $0 \le \Gamma \le 1$, where $\Gamma$ is the nominal amount of association. Therefore, Lee [35] solves numerically the equation

$$\gamma[\mathbf{p}(\lambda)] = \Gamma \qquad (40)$$

with respect to $\lambda$. With solution $\lambda^*$, the table $\mathbf{p}^\Gamma = \lambda^* \mathbf{p}^0 + (1-\lambda^*) \mathbf{p}^{\mathrm{opt}}$ satisfies the nominal $\Gamma$.

The main aim is to generate an $I_1 \times I_2 \times \cdots \times I_c$ table for $c$ categorical variables with given one-way marginal totals and nominal pairwise associations $\Gamma_{i,j}$, $i,j \in \{1,2,\cdots,c\}$, $i<j$. For each $\Gamma_{i,j}$, routine (40) is applied, leading to $c(c-1)/2$ two-way marginal totals $\mathbf{p}^{\Gamma_{i,j}}$. Each entry $p_{i',j'}^{\Gamma_{i,j}}$, with $i' \in \{1,2,\cdots,I_i\}$ and $j' \in \{1,2,\cdots,I_j\}$, can be expressed as a sum of the $c$–way probabilities, thus exhibiting linear equations. Lee [35] acknowledged that a solution of a system of linear equations with additional inequalities, $p_i \ge 0$, can be found by applying linear programming. Having determined an admissible table, the simulation is carried out with the inversion algorithm.

The described method of determining an admissible table will be called the $\gamma$–method from here on.

Neither Lee [35] nor Ibrahim and Suliadi [36] mentioned any problems finding a solution and gave the impression that the procedure always finds one. An example is given to prove that this is not always the case.

Consider a $2 \times 2 \times 2$ table with given one-way marginal totals $(0.2, 0.8)\prime$, $(0.4. 0.6)\prime$, and $(0.5, 0.5)\prime$ for variables one, two, and three, respectively. It can be confirmed that a table exists that satisfies the nominal pairwise association parameters $1 = \Gamma_{1,2} = \Gamma_{1,3} = \Gamma_{2,3}$. Now, the nominal pairwise association parameters are set to $\Gamma_{1,2} = -1$, $\Gamma_{1,3} = 1$, and $\Gamma_{2,3} = 1$. Routine (39) delivers the probabilities for the three $2 \times 2$ sub-tables $\mathbf{p}^{\Gamma_{1,2}}$, $\mathbf{p}^{\Gamma_{1,3}}$, and $\mathbf{p}^{\Gamma_{2,3}}$.

| $\Gamma_{1,2} = -1$ | | $\Gamma_{1,3} = 1$ | | $\Gamma_{2,3} = 1$ | | |
|---|---|---|---|---|---|---|
| 0 | 0.2 | 0.2 | 0 | 0.4 | 0 | (41) |
| 0.4 | 0.4 | 0.3 | 0.5 | 0.1 | 0.5 | |

It is not necessary to solve (40), since a priori, $\lambda = 1$ holds.

Now, the zero-, one-, and two-way marginal totals are known, and the system of linear equations can be established. There is one free parameter, and the three-way table satisfying the restraints is presented in Table 3.

From $p_{1,1,2} = -p_{1,1,1}$ and $p_{1,1,1}, p_{1,1,2} \ge 0$, it follows that $p_{1,1,1} = p_{1,1,2} = 0$ must hold. Therefore, $p_{2,2,1} = -0.1$ would follow; i.e., there is no table satisfying the restraints.

One could think that, if an admissible table exists, it can be determined by the $\gamma$-method. We now show that this is not correct. As an example, our task is to generate a $3 \times 3 \times 3$ table with one-way marginal totals $\mathbf{p}_1 = (0.1, 0.3, 0.6)\prime$, $\mathbf{p}_2 = (0.2, 0.4, 0.4)\prime$, and $\mathbf{p}_3 = (0.3, 0.3, 0.4)\prime$ and

**Table 3. A: Cell probabilities for given one- and two-way marginal totals.** B: Table for association parameters $\Gamma_{1,2} = -1$, $\Gamma_{1,3} = 1$, and $\Gamma_{2,3} = 0.714$.

| | | A) Bona fide table satisfying the restraints | | | | B) Table for extreme associations | | |
|---|---|---|---|---|---|---|---|---|
| | | $j = 1$ | $j = 2$ | | | | $j = 1$ | $j = 2$ |
| $i = 1$ | $k = 1$ | $p_{1,1,1}$ | $0.2-p_{1,1,1}$ | $i = 1$ | $k = 1$ | | 0 | 0.2 |
| | $k = 2$ | $-p_{1,1,1}$ | $p_{1,1,1}$ | | $k = 2$ | | 0 | 0 |
| $i = 2$ | $k = 1$ | $0.4-p_{1,1,1}$ | $p_{1,1,1}-0.1$ | $i = 2$ | $k = 1$ | | 0.3 | 0 |
| | $k = 2$ | $p_{1,1,1}$ | $0.5-p_{1,1,1}$ | | $k = 2$ | | 0.1 | 0.4 |

pairwise Goodman and Kruskal's association parameters $-\Gamma_{1,2} = \Gamma_{1,3} = \Gamma_{2,3} = 0.6023$. The sub-tables with maximum associations are determined via (39) and presented in Table 4.

The 3×3 sub-tables for independent variables were determined and are presented in Table 5.

Now, the 3×3 sub-tables for association parameters $-\Gamma_{1,2} = \Gamma_{1,3} = \Gamma_{2,3} = 0.6023$ are generated by determining the coefficient $\lambda$ due to Eq (40). The results are given in Table 6.

These two-way marginal totals are written as a linear system. Together with the inequalities $p_{i,j,k} \geq 0$, they should be solved by linear programming. As it turns out in this case, there is no solution. To see why, the linear system is solved to reduce the number of variables. From 3×3×3 variables $p_{i,j,k}$, there are eight free variables. It is not necessary to present the complete table. Five cell probabilities are the following:

$$p_{1,1,3} = 0.0107 - p_{1,1,1} - p_{1,1,2}$$

$$p_{1,2,3} = 0.0213 - p_{1,2,1} - p_{1,2,2}$$

$$p_{2,1,3} = 0.0320 - p_{2,1,1} - p_{2,1,2}$$

$$p_{2,2,3} = 0.0640 - p_{2,2,1} - p_{2,2,2}$$

$$p_{3,3,1} = p_{1,1,1} + p_{1,2,1} + p_{2,1,1} + p_{2,2,1} - 0.1502.$$

From the first four equations, it follows that $p_{1,1,1} \leq 0.0107$, $p_{1,2,1} \leq 0.0213$, $p_{2,1,1} \leq 0.032$, and $p_{2,2,1} \leq 0.064$. Hence, the sum $p_{1,1,1} + p_{1,2,1} + p_{2,1,1} + p_{2,2,1}$ is less than or equal to 0.128. Then, $p_{3,3,1}$, given by the last equation, is smaller than zero. Therefore, the $\gamma$-method is not able to find a solution for the formulated task.

However, there is a table satisfying the conditions that was found with the procedure NMaximize from Mathematica. The variable $\Gamma$ was maximized under the restraints of the zero- and one-way marginal totals $\Gamma = -\Gamma_{1,2} = \Gamma_{1,3} = \Gamma_{2,3}$ and nonnegative variables. The maximum was $\Gamma = 0.6023$, and the obtained table is given in Table 7.

To simplify the check of the side conditions, the sub-tables are given in Table 8. Although there are similarities to the two-way sub-tables in Table 4, there is one specific difference: zeros do appear, supporting an extreme table.

**Table 4. 3×3 sub-tables for association parameters $\Gamma_{1,2} = -1$, $\Gamma_{1,3} = 1$, and $\Gamma_{2,3} = 1$.**

| $\Gamma_{1,2} = -1$ | | | $\Gamma_{1,3} = 1$ | | | $\Gamma_{2,3} = 1$ | | |
|---|---|---|---|---|---|---|---|---|
| 0 | 0 | 0.1 | 0.1 | 0 | 0 | 0.2 | 0 | 0 |
| 0 | 0 | 0.3 | 0.2 | 0.1 | 0 | 0.1 | 0.3 | 0 |
| 0.2 | 0.4 | 0 | 0 | 0.2 | 0.4 | 0 | 0 | 0.4 |

**Table 5. 3×3 sub-tables for independent variables.**

| $\Gamma_{1,2} = 0$ | | | $\Gamma_{1,3} = 0$ | | | $\Gamma_{2,3} = 0$ | | |
|---|---|---|---|---|---|---|---|---|
| 0.02 | 0.04 | 0.04 | 0.03 | 0.03 | 0.04 | 0.06 | 0.06 | 0.08 |
| 0.06 | 0.12 | 0.12 | 0.09 | 0.09 | 0.12 | 0.12 | 0.12 | 0.16 |
| 0.12 | 0.24 | 0.24 | 0.18 | 0.18 | 0.24 | 0.12 | 0.12 | 0.16 |

The tool to determine an admissible association parameter scenario is still applied to the 2×2×2 table from above. Since there was no solution for $-\Gamma_{1,2} = \Gamma_{1,3} = \Gamma_{2,3} = 1$, it would be interesting to find an extreme constellation for which a solution would exist. The term $-\Gamma_{1,2}+\Gamma_{1,3}+\Gamma_{2,3}$ was maximized under the restraints of the one-way marginal totals and non-negative variables using the procedure NMaximize from Mathematica. The maximum was $-\Gamma_{1,2}+\Gamma_{1,3}+\Gamma_{2,3} = 2.714$ and the obtained table is given in Table 3B. The determination of the sub-tables and the comparison with the sub-tables of Table 2 shows that the first two sub-tables agree, but the third differs.

| $\Gamma_{2,3} = 0.714$ | |
|---|---|
| 0.3 | 0.1 |
| 0.2 | 0.4 |

The association parameter turned out to be 0.714. Therefore, it is possible to simulate a table for $-\Gamma_{1,2} = \Gamma_{1,3} = 1$ and $\Gamma_{2,3} = 0.714$. To simulate a table with agreeing association parameters (absolute values), we can determine an admissible table by applying NMaximize with the restraints $\Gamma = -\Gamma_{1,2} = \Gamma_{1,3} = \Gamma_{2,3}$ and maximize $\Gamma$. In this case, we obtain $\Gamma = 0.859$. If we infer the restraints $\Gamma = -\Gamma_{1,2} = \Gamma_{1,3} = \Gamma_{2,3}$, we obtain the admissible interval $-0.859 \leq \Gamma \leq 1$.

## 6.3 Association measured by Somers' *d*

There is one similarity between Pearson's $\rho$ and Goodman and Kruskal's $\gamma$: both take values between $-1$ and 1. A difference is that $\rho = 1$ means determinism, i.e., the observation of the category of one variable of an object is sufficient to know the category of the second variable.

| $\gamma = 1$ | | $\gamma = 0$ | | $\gamma = -1$ | |
|---|---|---|---|---|---|
| 0.1 | 0.7 | 0.08 | 0.72 | 0 | 0.8 |
| 0 | 0.2 | 0.02 | 0.18 | 0.1 | 0.1 |

This is not generally true for $\gamma = \pm 1$, since such an event only indicates that the table with maximum or minimum association is present. In fact, Lee [35] called it misleading perfect (negative) association. The three 2×2 tables shown here for $\gamma = 1, 0, -1$ have the same one-way marginals, $\mathbf{p}_1 = (0.8, 0.2)'$ and $\mathbf{p}_2 = (0.1, 0.9)'$.

**Table 6. 3×3 sub-tables for association parameters $-\Gamma_{1,2} = \Gamma_{1,3} = \Gamma_{2,3} = 0.6023$.**

| $\Gamma_{1,2} = -0.6023$ | | | $\Gamma_{1,3} = 0.6023$ | | | $\Gamma_{2,3} = 0.6023$ | | |
|---|---|---|---|---|---|---|---|---|
| $\lambda = 0.4670$ | | | $\lambda = 0.5100$ | | | $\lambda = 0.4872$ | | |
| 0.0107 | 0.0213 | 0.0680 | 0.0657 | 0.0147 | 0.0196 | 0.1282 | 0.0308 | 0.0410 |
| 0.0320 | 0.0640 | 0.2041 | 0.1461 | 0.0951 | 0.0588 | 0.1103 | 0.2077 | 0.0821 |
| 0.1574 | 0.3147 | 0.1279 | 0.0882 | 0.1902 | 0.3216 | 0.0615 | 0.0616 | 0.2769 |

**Table 7. 3×3×3 table satisfying $0.6023 = -\Gamma_{1,2} = \Gamma_{1,3} = \Gamma_{2,3}$ and one-way marginal totals $p_1 = (0.1, 0.3, 0.6)'$, $p_2 = (0.2, 0.4, 0.4)'$, and $p_3 = (0.3, 0.3, 0.4)'$.**

|  |  | $j = 1$ | $j = 2$ | $j = 3$ |
|---|---|---|---|---|
| $i = 1$ | $k = 1$ | 0 | 0.0262 | 0.0417 |
|  | $k = 2$ | 0 | 0 | 0.0321 |
|  | $k = 3$ | 0 | 0 | 0 |
| $i = 2$ | $k = 1$ | 0.0072 | 0.1193 | 0 |
|  | $k = 2$ | 0 | 0.0181 | 0.0820 |
|  | $k = 3$ | 0 | 0 | 0.0738 |
| $i = 3$ | $k = 1$ | 0.1056 | 0 | 0 |
|  | $k = 2$ | 0.0872 | 0.0806 | 0 |
|  | $k = 3$ | 0 | 0.1558 | 0.1708 |

The respective Pearson's correlation coefficients are $\rho = 0.1667$, $\rho = 0$, and $\rho = -0.6667$. That means, for the given one-way marginals, that $\gamma = 1$ stands for low (positive) association, while $\gamma = -1$ stands for large (negative) association. Hence, the $\gamma$-scale is a relative one and worthless without additional information. Following [40], Somers' $d$ is a better measure of association (dependence) between ordinal variables. It is a modification of Goodman and Kruskal's $\gamma$. Since a symmetrical version is needed here, the definition of $T$ becomes

$$T = \sum_{i<i'}\sum_{j=j'} p_{i,j}p_{i',j'} + \sum_{i=i'}\sum_{j<j'} p_{i,j}p_{i',j'} + \sum_{i=i'}\sum_{j>j'} p_{i,j}p_{i',j'} + \sum_{i>i'}\sum_{j=j'} p_{i,j}p_{i',j'} \qquad (42)$$

and the definition of $d$ is

$$d = \frac{S - D}{S + D + T} = \frac{S - D}{1 - \sum_i\sum_j p_{i,j}^2}. \qquad (43)$$

The right-hand-side version results from $1 = \sum_i\sum_j\sum_{i'}\sum_{j'} p_{i,j}p_{i',j'}$ and $S + D + T = 1 - \sum_{i=i'}\sum_{j=j'} p_{i,j}p_{i',j'} = 1 - \sum_i\sum_j p_{i,j}^2$.

It is easy to see that $d = 0$ holds if the variables are independent, and $d = \pm 1$ holds if the category of one variable can be deduced from knowing the category of the other variable, i.e., when the table has a (anti-) diagonal structure. For the 2×2 tables from above, $d = 0.087$, $d = 0$, and $d = -0.471$ hold, respectively.

To give an impression of the relation of Somers' $d$ and Pearson's $\rho$, the parameters were calculated for the sub-tables of Table 8. Somers' values were $d_{1,2} = -0.686$, $d_{1,3} = 0.622$, and $d_{2,3} = 0.857$, and Pearson's values were $\rho_{1,2} = -0.797$, $\rho_{1,3} = -0.808$, and $\rho_{2,3} = 0.933$.

To find an admissible table satisfying the one-way marginals and the nominal pairwise association parameters $\Delta_{i,j}$, it is possible to apply a slightly modified version of the $\gamma$−method. For a certain pair $i,j$ of variables, $\mathbf{p}^0$ and $\mathbf{p}^{\text{opt}}$ (which are $\mathbf{p}^{\text{max}}$ for $\Delta > 0$ and $\mathbf{p}^{\text{min}}$ for $\Delta < 0$) are determined as before. It is useful to calculate $d$ for the table $\mathbf{p}^{\text{opt}}$. The nominal $\Delta$ should reflect less

**Table 8. Two-way sub-tables from Table 7.**

| $\Gamma_{1,2} = -0.6023$ | | | $\Gamma_{1,3} = 0.6023$ | | | $\Gamma_{2,3} = 0.6023$ | | |
|---|---|---|---|---|---|---|---|---|
| 0 | 0.0262 | 0.0738 | 0.0679 | 0.0321 | 0 | 0.1128 | 0.0872 | 0 |
| 0.0072 | 0.1374 | 0.1554 | 0.1265 | 0.1001 | 0.0734 | 0.1454 | 0.0987 | 0.1558 |
| 0.1928 | 0.2364 | 0.1708 | 0.1056 | 0.1678 | 0.3266 | 0.0417 | 0.1141 | 0.2442 |

association than $d$. Then, similar to the $\gamma$-method, for each pair of variables,

$$\mathbf{p}(\lambda) = (1 - \lambda)\mathbf{p}^0 + \lambda\mathbf{p}^{\mathrm{opt}}$$
$$d[\mathbf{p}(\lambda)] = \Delta \tag{44}$$

must be solved numerically. As with the $\gamma$–method, the obtained two-way marginals $\mathbf{p}(\lambda^*)$ are written as a system of linear equations. These are solved by linear programming software. If no solution is obtained, the nominal association parameters need to be weakened.

This was the analog to the $\gamma$-method. The additional tools presented in Sections 6.1 and 6.2 can be adapted.

Consider again the example of the 3×3×3 table with one-way marginals $\mathbf{p}_1 = (0.1, 0.3, 0.6)'$, $\mathbf{p}_2 = (0.2, 0.4, 0.4)'$, and $\mathbf{p}_3 = (0.3, 0.3, 0.4)'$. Then, the bounds $-0.686 \leq d_{1,2} \leq 0.595$, $-0.667 \leq d_{1,3} \leq 0.622$, and $-0.667 \leq d_{2,3} \leq 0.857$ are obtained. Maximizing $d_{1,2}+d_{1,3}+d_{2,3}$ yields 2.074, with $d_{1,2} = 0.594$, $d_{1,3} = 0.622$, and $d_{2,3} = 0.857$. Minimizing $d_{1,2}+d_{1,3}+d_{2,3}$ yields $-0.984$, with $d_{1,2} = -0.244$, $d_{1,3} = -0.073$, and $d_{2,3} = -0.667$. For $d = d_{1,2} = d_{1,3} = d_{2,3}$, the admissible interval is $-0.308 \leq d \leq 0.594$.

For the example of Table 3B, $d_{1,2} = -0.250$, $d_{1,3} = 0.323$, and $d_{2,3} = 0.286$ are calculated. (For comparison, Pearson's correlation coefficients were $\rho_{1,2} = -0.408$, $\rho_{1,3} = 0.500$, and $\rho_{2,3} = 0.408$.)

For the sub-tables of Table 8, $d_{1,2} = -0.255$, $d_{1,3} = 0.282$, and $d_{2,3} = 0.318$ are calculated. (For comparison, Pearson's correlation coefficients were $\rho_{1,2} = -0.390$, $\rho_{1,3} = 0.429$, and $\rho_{2,3} = 0.475$.)

## 6.4 How to obtain tables for all admissible associations measured by Somers' d

As was worked out in the last section, the adapted $\gamma$–method does not always allow the determination of a table satisfying nominal associations measured by Somers' $d$, although one exists. It was also reported that a numerical maximization was able to find the solution. However, a large number of iterations were necessary, and the method may fail if the number of variables increases.

Assume that a table $\mathbf{p}^*$ exists that satisfies the nominal two-way associations measured by Somers' $d$. Let the expression $\mathbf{d}^* = d(\mathbf{p}^*)$ define this property, where $\mathbf{d}^*$ is the vector of the nominal two-way associations and $d(\mathbf{p})$ indicates the vector of the actual two-way associations from table $\mathbf{p}$.

From Section 6.1, it is known how to determine a table with given Pearson's correlation coefficients $\boldsymbol{\rho}$, where $\boldsymbol{\rho}$ is the vector of the two-way correlation coefficients. Denote the generated table with $\mathbf{p}(\boldsymbol{\rho})$. We are looking now for $\boldsymbol{\rho}$, so that $\mathbf{p}(\boldsymbol{\rho})$ has the desired property with respect to $\mathbf{d}^*$, i.e., $\mathbf{d}^* = d[\mathbf{p}(\boldsymbol{\rho})]$. This is realized by a minimization procedure:

$$\min_{\boldsymbol{\rho}} \ \|\mathbf{d}^* - d[\mathbf{p}(\boldsymbol{\rho})]\|. \tag{45}$$

We used the function FindMinimum of Mathematica with starting points $\boldsymbol{\rho} = \mathbf{d}^*$ and the Euclidean norm. The function $\mathbf{p}(\boldsymbol{\rho})$ had to be specified in two ways, and subsequently, the minima and maxima of the free variables were evaluated. The cell of the actual variable was set to the mean of the minimum and maximum. We denote the specification with $\mathbf{p}(\boldsymbol{\rho}, \mathrm{mean})$. When $\boldsymbol{\rho}$ left the admissible region, i.e., when there was no solution for the restraints, the penalty term $\|\mathbf{d}^* - d[\mathbf{p}(\boldsymbol{\rho}, \mathrm{mean})]\|$ was set to a large value. The argument for which the norm is minimum is named $\boldsymbol{\rho}^*$.

The procedure was applied to the repeatedly used one-way marginals of a 3×3×3 table. In Section 6.3, the admissible range $-0.308 \leq d \leq 0.594$ was determined for $d = d_{1,2} = d_{1,3} = d_{2,3}$. Nearly extreme scenarios have been chosen. For $\mathbf{d}^* = (d_{1,2}^*, d_{1,3}^*, d_{2,3}^*)' = (-0.3, -0.3, -0.3)\prime$,

the appropriate $\boldsymbol{\rho}^*$ vector became $(-0.437, -0.447, -0.456)'$. The obtained table is not presented here but can be determined via $\mathbf{p}(\boldsymbol{\rho}^*, \text{mean})$ and the technique from Section 6.1. For $\mathbf{d}^* = (0.59, 0.59, 0.59)'$, the appropriate $\boldsymbol{\rho}^*$ vector became $(0.791, 0.764, 0.750)'$. It was confirmed with other examples that the Pearson's coefficients were often (absolutely) larger the Somers'. But this is not a general rule, as proven with $\mathbf{d}^* = (0, 0, 0)'$. Then, the appropriate $\boldsymbol{\rho}^*$ vector became $(0.182, 0.057, 0.013)'$ and deviated considerably from the expected $\boldsymbol{\rho}^* \approx (0, 0, 0)'$.

Recall that a solution need not be unique. Assuming independence between all pairs of variables, the related table is determined by multiplying the one-way marginals involved in the specified cells. For this independence table, $\mathbf{d} = (0, 0, 0)'$ and $\boldsymbol{\rho} = (0, 0, 0)'$ hold. If we wish to generate the independence table, given the demand $\mathbf{d}^* = (0, 0, 0)'$, we must give up the choice of the mean value of the admissible intervals $(b^l, b^u)$ of the free variables. Instead, we take that value of the interval that is nearest to the value $p^{\text{ind}}$ of the independence table. If $b^l \le p^{\text{ind}} \le b^u$ holds, $p = p^{\text{ind}}$ is taken, and if $p^{\text{ind}} < b^l$ holds, $p = b^l$ is taken. For $b^u < p^{\text{ind}}$, the choice is $p = b^u$. We denote this specification with $\mathbf{p}(\boldsymbol{\rho}, \text{ind})$. The application of this principle led indeed to $\boldsymbol{\rho}^* = (0, 0, 0)'$. Applied to $\mathbf{d}^* = (-0.3, -0.3, -0.3)'$, the appropriate $\boldsymbol{\rho}^*$ vector became $(-0.442, -0.440, -0.454)'$. For $\mathbf{d}^* = (0.59, 0.59, 0.59)'$, the appropriate $\boldsymbol{\rho}^*$ vector became $(0.790, 0.750, 0.735)'$. Obviously, for high associations, the difference between $\mathbf{p}(\boldsymbol{\rho}, \text{ind})$ and $\mathbf{p}(\boldsymbol{\rho}, \text{mean})$ was not great. For the most extreme associations, $-0.308$ and $0.594$, $\mathbf{p}(\boldsymbol{\rho}, \text{ind})$ and $\mathbf{p}(\boldsymbol{\rho}, \text{mean})$ result in the same table.

For $\mathbf{d}^* = (0, 0, 0)'$, $\mathbf{p}(\boldsymbol{\rho}, \text{max})$ was still evaluated, i.e., the maximum was always chosen from the admissible intervals for the free variables. Then, the appropriate $\boldsymbol{\rho}^*$ vector became $(0.193, 0.130, 0.026)'$. Analogously, with $\mathbf{p}(\boldsymbol{\rho}, \text{min})$, the appropriate $\boldsymbol{\rho}^*$ vector became $(-0.100, -0.090, -0.086)'$. This might suffice to illustrate the admissible range of tables satisfying nominal associations.

When the minimum of (45) was not zero for a nominal $\mathbf{d}^*$, no table for the demands was found. Then, the nearest admissible table due to the used norm was obtained.

## 7 Application to the Berkeley data

### 7.1 Why the two-way LD differs from the partial LDs and their mean

One real-life example for Simpson's paradox is particularly impressive. The University of California, Berkeley, was sued for bias against women who had applied for admission. The reduced data version found at https://en.wikipedia.org/wiki/Simpson%27s_paradox is presented in columns 1–5 of Table 9.

**Table 9. Numbers of denied and admitted applications at six departments as part of the study [41].** Variable 1 is sex (men—women), variable 2 is admittance (denied–admitted), and variable 3 is the department (1 to 6). $D_{1,3_i}$ is the LD between variable 1 and variable 3 (which is now dichotomous: Department $i$ versus the rest). $D_{2,3_i}$ is the LD between variable 2 and variable 3 (which is again dichotomous: Department $i$ versus the rest). Parameter $p_{3_i}$ stands for the frequency of applications to department $i$. $D_{1,2|3_i}$ is the LD between the first category of variable 1 and the first category of variable 2 within Department $i$, and $\rho_{1,2|3_i}$ is the corresponding correlation coefficient.

| Dept. | Men | | Women | | $D_{1,3_i}$ | $D_{2,3_i}$ | $p_{3_i}$ | $D_{1,2|3_i}$ | $\rho_{1,2|3_i}$ |
|---|---|---|---|---|---|---|---|---|---|
| $i$ | Denied | Admitted (%) | Denied | Admitted (%) | | | | | |
| 1 | 313 | 512 (62) | 19 | 89 (82) | 0.060 | −0.053 | 0.206 | 0.021 | 0.14 |
| 2 | 207 | 353 (63) | 8 | 17 (68) | 0.047 | −0.032 | 0.129 | 0.002 | 0.02 |
| 3 | 205 | 120 (37) | 391 | 202 (34) | −0.049 | 0.008 | 0.203 | −0.007 | −0.03 |
| 4 | 279 | 138 (33) | 244 | 131 (35) | −0.012 | 0.008 | 0.175 | 0.005 | 0.02 |
| 5 | 138 | 53 (28) | 299 | 94 (24) | −0.035 | 0.018 | 0.129 | −0.008 | −0.04 |
| 6 | 351 | (22 (6) | 317 | 24 (7) | −0.011 | 0.051 | 0.158 | 0.003 | 0.02 |
| Total | 1493 | 1198 (44.5) | 1278 | 557 (30.4) | 0 | 0 | 1 | 0.0034 | 0.03 |

Dividing the number of admitted men by the number of applying men shows a rate of 1198/2691 = 44.5%, while dividing the number of admitted women by the number of applying women shows a rate of 557/1835 = 30.4%. The large difference between 44.5% and 30.4% resulted in a perception of discrimination against women. Therefore, the question was whether women were really handicapped or if there were other reasons that led to the differing rates.

Bickel and collegues [41] examined the department-level data and did not find clear evidence of discrimination against women. Averaged over the departments, they found a moderate preference for women. In principle and qualitatively, this corresponds to the inspection of the last two columns of Table 9. Note that positive values mean that more men than women relative to their frequencies were denied; i.e., more women were admitted. The authors of [41] also worked out the reason for the great discrepancy between the apparent overall handicap for women and the almost absent handicaps within the departments. The reason was the preferred applications of women to departments with low admission rates. However, this reason was not found by straightforward theory but by good detective work.

With the LD approach of Section 2, the parameter for the overall association between sex and admission is $D_{1,2}$. The overall LD is $D_{1,2} = (1493 \times 557 - 1198 \times 1278)/4526^2 = -0.0341$, showing the handicap for women. (Significance tests should and can be applied, but they are not the focus here.) This approach does not account for the influence of the departments. Therefore, the averaged LDs of the departments $D_{1,2|3_i}$ is a more reliable parameter. Direct evaluation of $\bar{D}_{1,2|3}$ via the eighth column of Table 9 gives $\bar{D}_{1,2|3} = 0.0034$, reflecting a small preference for women. The difference between both parameters is $D_{1,2} - \bar{D}_{1,2|3} = -0.0375$.

Now, the result (8) of Section 2 is applied. With it, the difference can be determined in a completely different manner. The difference between $D_{1,2}$ and $\bar{D}_{1,2|3}$ is $\sum_{i=1}^{6} D_{1,3_i} D_{2,3_i}/p_{3_i}$. The first summand, $D_{1,3_1} D_{2,3_1}/p_{3_1}$, is determined as follows. $D_{1,3_1}$ belongs to the 2×2 table for Department 1, where the first column is assigned to "men" and the second column to "women". The first row is assigned to "Department $i$". The second row is assigned to the complement, i.e., to the rest of departments. In Department 1, 825 men and 108 woman applied. Overall, 2691 men and 1835 women applied; i.e., 2691−825 men and 1835−108 women applied to the other departments. The numbers are presented in Table 10 together with those for denied and admitted applicants at department 1.

We obtain $D_{1,3_1} = (825 \times 1727 - 108 \times 1866)/4526^2 = 0.056$. The positive sign says that the applications of men appeared more often than the average. Analogously, $D_{2,3_1} = (332 \times 1154 - 601 \times 2439)/4526^2 = -0.053$ is calculated. The negative sign indicates that admittance was more often than the average. We need still $p_{3_1}$, the probability of application to Department 1, which is $p_{3_1} = (825 + 108)/4526 = 0.206$. Hence, $D_{1,3_1} D_{2,3_1}/p_{3_1} = -0.056 \times 0.053/0.206 = -0.014$. When columns 6, 7, and 8 are completed, the difference between $D_{1,2}$ and $\bar{D}_{1,2|3}$ can be calculated:

$$D_{1,2} - \bar{D}_{1,2|3} = \sum_{k=1}^{6} \frac{D_{1,3_i} D_{2,3_i}}{p_{3_i}} = -0.0375. \tag{46}$$

**Table 10. Numbers of men and women with application to department 1 and numbers of denied and admitted applicants at department 1.** "Rest" means departments two to six.

|  | Men | Women | Denied | Admitted |
|---|---|---|---|---|
| Department 1 | 825 | 108 | 332 | 601 |
| Rest | 1866 | 1727 | 2439 | 1154 |
| Total | 2691 | 1835 | 2771 | 1755 |

It is easy to see that the difference becomes particularly large (positive) when $D_{1,3_i}$ and $D_{2,3_i}$ have the same sign within the departments, because then all summands are positive. Also, the difference becomes particularly small (negative) when $D_{1,3_i}$ and $D_{2,3_i}$ have different signs within the departments, because then all summands are negative. Inspection of columns 6 and 7 of Table 9 proves the negative correlation of $D_{1,3_i}$ and $D_{2,3_i}$ within the Berkeley data.

The interpretation is that the apparent discrimination against women was caused by a property of the departments. Those with high admittance rates had more male applicants and those with low admittance rates had more female applicants. While Bickel and colleagues [41] had to be good detectives to discover this trend, the new approach makes it obvious immediately. The remark in [41] concerning the role of the size of the departments has to be verified, since $p_{3_i}$ appears in the denominator in (46). See also Eq (49B).

## 7.2 The determination of parsimonious models fitting the data

The aim is to infer whether a proven three-way interaction is caused by all three-way interaction parameters or only by a subset of them. With the Berkeley data, it is shown that the search for a parsimonious model fitting the data can be successful.

The multinomial distribution of Table 9 has 2×2×6 = 24 parameters. Eliminating zero-, one-, and two-way marginals results in five free variables. To determine the distribution without a three-way interaction, the entropy was maximized for these variables. The maximum 2.888 was reached for $\tilde{p}_{1,1,1} = 0.0653$, $\tilde{p}_{1,1,2} = 0.0456$, $\tilde{p}_{1,1,3} = 0.0477$, $\tilde{p}_{1,1,4} = 0.0618$, and $\tilde{p}_{1,1,5} = 0.0321$. Comparison with the observed table yields a $\chi^2$ value of 18.8, speaking to non-agreement ($p = 0.002$) and the existence of a three-way interaction.

The three-way interaction is quantified by the three-way interaction parameters $D_{1,1,i} = n_{1,1,i}/4526 - \tilde{p}_{1,1,i}$. The counts for the variables were $n_{1,1,1} = 313$, $n_{1,1,2} = 207$, $n_{1,1,3} = 205$, $n_{1,1,4} = 279$, and $n_{1,1,5} = 138$. The corresponding three-way interaction parameters are therefore $D_{1,1,1} = 0.0038$, $D_{1,1,2} = 0.00014$, $D_{1,1,3} = -0.0024$, $D_{1,1,4} = -0.00018$, and $D_{1,1,5} = -0.0016$. The sixth three-way disequilibrium parameter, $D_{1,1,6}$, linearly depends on the others. Actually, the sum $\sum_{i=1}^{6} D_{1,1,i}$ is zero, i.e., $D_{1,1,6} = 0.00021$. The largest absolute value appeared for the first department.

All three-way interaction parameters differing from zero reflect a contribution to three-way interaction. To quantify these contributions, the partial 2×2 tables under the hypothesis of an absence of three-way interactions were compared with the observed ones. The $\chi^2$ values for the six categories were 20.6, 0.1, 2.4, 0.01, 2.1, and 0.1, respectively. Obviously, the first department indeed plays a dominant role.

A table $\{p_{i,j,k}\}$ fitting the observed table must therefore trim $D_{1,1,1}$ to zero. This can be guaranteed by setting the free parameter $p_{1,1,1}$ to $n_{1,1,1}/4526$. Then, there remain four free parameters. Theoretically, one could now derive maximum-likelihood estimates to fit them, but the use of the maximum entropy principle under restraints is easier. The restraints are the zero-, one-, and two-way marginals and $p_{1,1,1} = 0.0692$. Then, the four remaining free parameters are determined by numerically maximizing the entropy. The maximum $H = 2.886$ was reached for $p_{1,1,2} = 0.0454$, $p_{1,1,3} = 0.0463$, $p_{1,1,4} = 0.0605$, and $p_{1,1,5} = 0.0314$. A comparison of the corresponding table with the observed one yielded a $\chi^2$ value of 2.56; i.e., the data were met. For the hypothetical table, the partial correlations $\rho_{1,2|3_i}$ for Departments 1 to 6 were 0.136, −0.003, −0.007, −0.007, −0.006, and −0.004, respectively; i.e., with the exception of $\rho_{1,2|3_1}$, they were absolutely small. The complete table, multiplied by $n$, is presented in Table 11 under Method A.

**Table 11. Fitted counts of the Berkeley data.** Five free variables were fitted in three ways. A: First variable $n_{1,1,1}$ taken from Table 9, four from maximizing entropy. B: Five from $D_{1,1,i} = 0$, $i = 2,3,\ldots,6$. C: Four from agreeing $\rho_{1,2|3_i} = \rho_{1,2|3_{i+1}}$, $i = 2, 3, 4, 5$, with the fifth the log-likelihood estimate.

| Dep. | Men | | | | | | Women | | | | | |
|---|---|---|---|---|---|---|---|---|---|---|---|---|
| | Denied | | | Admitted | | | Denied | | | Admitted | | |
| $i$ | A | B | C | A | B | C | A | B | C | A | B | C |
| 1 | 313.0 | 308.9 | 312.7 | 512.0 | 516.1 | 512.3 | 19.0 | 23.1 | 19.3 | 89.0 | 84.9 | 88.7 |
| 2 | 205.6 | 205.8 | 205.5 | 354.4 | 354.2 | 354.5 | 9.4 | 9.2 | 9.5 | 15.6 | 15.8 | 15.5 |
| 3 | 209.5 | 211.0 | 209.8 | 115.5 | 114.0 | 115.2 | 386.5 | 385.0 | 386.2 | 206.5 | 208.0 | 206.8 |
| 4 | 274.0 | 275.4 | 274.3 | 143.0 | 141.6 | 142.7 | 249.0 | 247.6 | 248.7 | 126.0 | 127.4 | 126.3 |
| 5 | 142.2 | 142.9 | 142.2 | 48.8 | 48.1 | 48.8 | 294.8 | 294.1 | 294.8 | 98.2 | 98.9 | 98.2 |
| 6 | 348.6 | 349.0 | 348.5 | 24.4 | 24.0 | 24.5 | 319.4 | 319.0 | 319.5 | 21.6 | 22.0 | 21.5 |

Now an alternative method is investigated. In Section 3, we worked out for 2×2×2 tables how to infer the agreement of partial correlations. This approach is straightforward to generalize to 2×2×$I_3$ tables with $I_3 > 2$. (For that, the right-hand-side expression of Eq (14) is useful. There, $i$ substitutes for index "1" and $I_3$ substitutes for index "2". The same substitutions are necessary for the third indices of $A$ and $B$.) Then, the global hypothesis for the Berkeley data would be that all partial correlations with respect to the third variable agree. It can be formulized by demanding $\rho_{1,2|3_1} = \rho_{1,2|3_2} = \cdots = \rho_{1,2|3_6}$; i.e., there are actually five equations. Since there are also five degrees of freedom, the hypothetical table may be calculated explicitly. The hypothetical agreeing partial correlation coefficients became 0.019. A comparison of the hypothetical 2×2×6 table with the observed data gave a $\chi^2$ value of 17.4 ($p = 0.0036$); i.e., one cannot be convinced with unique correlation coefficients. Comparisons of the partial 2×2 tables with the observed ones showed one significant deviation. For Department 1, the $\chi^2$ value became 19.2. All other $\chi^2$ values were smaller than 2.1.

Comparisons of the partial 2×2 tables with the tables for independence showed no significant deviation. All $\chi^2$ values were smaller than 0.4. That means $0 = D_{1,2|3_2} = D_{1,2|3_3} = \cdots = D_{1,2|3_6}$ can be assumed. These five equations are solved by the five arguments $p_{1,1,1} = 0.0683$, $p_{1,1,2} = 0.0455$, $p_{1,1,3} = 0.0466$, $p_{1,1,4} = 0.0608$, and $p_{1,1,5} = 0.0316$. The comparison of the associated table with the observed data gave a $\chi^2$ value of 3.69; i.e., the model fits the data. Comparisons of the partial 2×2 tables with the observed ones also showed no significant deviation. All $\chi^2$ values were less than 1.2. Comparisons of the partial 2×2 tables with those under independence showed one significant deviation. For Department 1, the $\chi^2$ value became 10.8 ($p = 0.001$, the correlation was $\rho_{1,2|3_1} = 0.107$, somewhat smaller than before). All other $\chi^2$ values were of course zero. The table is presented under Method B in Table 11.

Method B gained from the finding that five partial correlations could be set to zero. Under different circumstances, it could be possible that the five partial correlations agree but are not zero. In that case, $\rho_{1,2|3_i} = \rho_{1,2|3_{i+1}}$ can be assumed for $i \in \{2, 3, 4, 5\}$. For these four equations, four variables can be eliminated. The last variable, $p_{1,1,1}$, is determined via maximum likelihood, i.e., by maximizing $\Sigma_{i,j,k} n_{i,j,k} \ln(p_{i,j,k})$. The solution can be viewed under Method C in Table 11. The comparison of the table with the observed data gave a $\chi^2$ value of 2.73; i.e., the model fits the data. Comparisons of the partial 2×2 tables with the observed ones also showed no significant deviation. All $\chi^2$ values were less than 0.72. Comparisons of the partial 2×2 tables with those under independence showed one significant deviation. For Department 1, the $\chi^2$ value became 16.7 ($p = 0.00004$, the correlation was $\rho_{1,2|3_1} = 0.134$). All other $\chi^2$ values were less than 0.03.

Thus, the apparent discrimination against women with respect to admittance turned out to be untrue. In Departments 2 to 6, men and women were admitted equally. In the first department, men had a significant handicap.

## 8 Discussion

In Section 2, the LD parameter was used to quantify Simpson's paradox. The difference between a two-way interaction and the averaged partial interactions was derived. For a 2×2×2 table, the difference was

$$D_{1,2} - \bar{D}_{1,2|3} = \frac{D_{1,3}D_{2,3}}{p_3(1-p_3)}. \tag{47}$$

(Note that the notation $D_{1,2}$, e.g., means the LD between the first category of variable 1 and the first category of variable 2, formerly denoted by $D_{1_1,2_1}$.) In many experiments, there is one response variable (here it is the first one) and two explanatory variables. The latter ones can be arranged to ensure $D_{2,3} = 0$, for example, by applying the treatments to the same fraction of males and females. In this way, the difference is zero and Simpson's paradox is circumvented.

Let $D_{1,3}$ and $D_{2,3}$ differ from zero. One could think that the difference is largest when $p_3$ is near zero or one. However, the value of LD depends on the one-way marginal totals. Using the correlation coefficients (5) instead gives

$$D_{1,2} - \bar{D}_{1,2|3} = \sqrt{p_1(1-p_1)p_2(1-p_2)}\rho_{1,3}\,\rho_{2,3}. \tag{48}$$

Thus, the absolute difference is largest when $p_1$ and $p_2$ are one half, and it is smallest when $p_1$ or $p_2$ are zero or one. On the other hand, when $p_1$ and $p_2$ are zero or one, the associated LD, i.e., $D_{1,2}$, is zero. Therefore, it is useful to also consider the relative difference:

$$\frac{D_{1,2} - \bar{D}_{1,2|3}}{D_{1,2}} = \frac{\sqrt{p_1(1-p_1)p_2(1-p_2)}\rho_{1,3}\,\rho_{2,3}}{D_{1,2}} = \frac{\rho_{1,3}\,\rho_{2,3}}{\rho_{1,2}}. \tag{49A}$$

For an $I_1 \times I_2 \times I_3$ table, the appropriate expression is

$$1 - \frac{\bar{D}_{1_i,2_j|3}}{D_{1_i,2_j}} = \sum_{k=1}^{I_3}(1-p_{3_k})\frac{\rho_{1_i,3_k}\rho_{2_j,3_k}}{\rho_{1_i,2_j}}. \tag{49B}$$

When we are interested in the association between two categorical variables, such as sex and admission at a university, it is useful to determine $D_{1,2}$ or $\rho_{1,2}$. If one finds preference for one sex, this does not mean that the other sex experienced discrimination. The reason could be that the abilities of the sexes happened to be different. Therefore, it was reasonable to consider an index for the high school report as a factor. With the Berkeley data, it turned out that the departments need to be considered as a factor. When a third factor has an effect, then $\bar{D}_{1,2|3}$ gives a better estimate for the association of the two variables than $D_{1,2}$. However, the value of $D_{1,2}$ is still useful. If the difference $D_{1,2} - \bar{D}_{1,2|3}$ is greater than zero, it follows automatically from (47) that $D_{1,3}$ and $D_{2,3}$ cannot be zero and they have the same signs. If the difference is zero, it follows that $D_{1,3}$ or $D_{2,3}$ are zero. If the difference is smaller than zero, $D_{1,3}$ and $D_{2,3}$ cannot be zero and they have different signs; i.e., the interactions have different directions.

Hence, Eqs (8) and (9) are a great help for interpreting the tables. It is particularly interesting that their validity is independent of the free parameters.

Section 3 was dedicated to the question of whether the amount of interaction between two variables depends on another categorical variable. In Section 7.2, the approach was generalized to $2 \times 2 \times I_3$ tables and applied to the Berkeley data. It was possible to find parsimonious models that fit the data.

If all variables have more than two categories, i.e., for a general $I_1 \times I_2 \times I_3$ table, the third variable has no effect on the associations between the first and second variable if

$$\rho_{1_i, 2_j | 3_k} = \rho_{1_{i+1}, 2_{j+1} | 3_k} \tag{50}$$

holds for $i = 1, 2, \cdots, I_1-1, j = 1, 2, \cdots, I_2-1$, and $k = 1, 2, \cdots, I_3-1$. Analogously to (12), (13), and (14), this system of linear equations can be solved, thereby delivering the hypothetical table that can be compared with the observed one. If it does not fit the data, subsequently it can be checked whether the $k$–th category of the third variable plays a special role. For each $k$, the $(I_1-1)(I_2-1)$ equations of (50) are solved. The remaining $(I_1-1)(I_2-1)(I_3-2)$ free variables are found by maximizing entropy. For each $k$, the hypothetical table can again be tested against the observed table.

When there is still no hit, the largest deviations from an average $\rho$ can be searched in different ways. There is a need for further investigations into an optimal systematic strategy to find a parsimonious model. The model choice and multiple testing theories have to be kept in mind.

However, the new approach is suited to answering important questions and surely enriches the theory of contingency tables.

In Section 4, the relation between Bartlett's and Bennett's measure on the three-way interaction was investigated. As summarized in the introduction, Bartlett's measure (which he mentioned came from R.A. Fisher) had a high degree of impact, while Bennett's measure was a generalization of the two-way LD based on intuition. The meaning and correctness of this measure could therefore only be checked through its relation to Bartlett's measure. As it turned out, it is a simplified version of the first-order Taylor expansion of the latter one.

For $2 \times 2 \times 2 \times 2$ tables, the criterion for an absence of four-way interaction is a straightforward generalization of Bartlett's multiplicative criterion, shown by Good [27]. A generalization of Bennett's linear three-way measure is not straightforward [16–20].

Unfortunately, the roots of the seven-degree polynomial arising for the multiplicative measure cannot be determined algebraically, and the Taylor expansion cannot be generated directly. However, the criterion is a function of the parameter $p_{1,1,1,1}$, which is itself a function of the one-, two-, and three-way marginal totals. Thus, further progress depends on the availability of an effective algorithm to derive multivariate Taylor expansions for implicit functions. Since the implicit function is a polynomial (where the order of derivatives unequal to zero is finite) and there is a high amount of symmetry, there appears to be hope.

The focus in Section 5 was on tables with zero counts. The question was whether these counts appeared by chance or whether they were a necessary consequence from the given two-way marginal totals. The application of linear programming was successful in obtaining fixed zero counts. Furthermore, fixed nonzero counts can be determined.

One example of Fienberg and Rinaldo [2] was reanalyzed and lead to Table 2. For completeness, the other examples were also investigated. For their Tables 4 and 7, no fixed cells were obtained. For their Table 5, all cells turned out to be fixed. Fienberg and Rinaldo [2] characterized this table as yielding no MLE and wrote, "In fact, the values of both goodness of fit statistics will always be almost zero, *no matter what the positive counts are*". This underlines that they did not acknowledge that the contingency table was the only one with the given marginal totals. So far, there was no tool available to find this simple but important truth.

In cases where the number of variables could be reduced, such as with Table 2, the question arises whether the determination of the MLEs can be optimized. One way would be to modify commonly used procedures. Alternatively, Good's [27] method of maximizing the entropy under restraints can be used. Numerical maximization of the entropy of the data of Table 2 (divided by $n = 113$), given the two-way marginal totals, yielded $p_{1,1,2} = 0.0316$, $p_{1,1,3} = 0.0120$,

$p_{2,3,1} = 0.0102$, and $p_{3,2,1} = 0.0081$. Due to the concavity of entropy, the convergence was excellent. The other cell counts can be calculated according to the expressions in Table 2.

In Section 6, improvements were achieved for the simulation of ordinally scaled variables. The main task was to determine an admissible table satisfying the demands. As noted above, there might be difficulties in obtaining an admissible table, simply because such a table does not exist when the restraints are too strong. Ignoring the assumptions $p_i \geq 0$, a solution would exist (if the number of equations expressing the restraints does not exceed the number of variables). However, the bona fide table would have negative entries. Therefore, one can search for an admissible solution by minimizing

$$\sum_{\substack{i = 1 \\ p_i < 0}}^{\prod_{j=1}^{c} I_j} p_i^2, \tag{51}$$

where the free variables must be fit. If the obtained minimum is zero, an admissible solution is found. Alternatively, the maximum entropy principle under restraints can be used. When a bona fide table has negative entries, the entropy becomes complex. Therefore, it is useful to minimize

$$\sum_{i=1}^{\prod_{j=1}^{c} I_j} \{\mathrm{Im}[H(\mathbf{p})]\}^2, \tag{52}$$

where $\mathrm{Im}(x)$ is the imaginary part of $x$. Our limited experiences would emphasize the latter method.

Lee [35] derived an algorithm for simulating nominal variables with given pairwise correlations measured with Goodman and Kruskal's $\tau$, $0 \leq \tau \leq 1$. Since the original measure does not ensure $\tau(Y_1, Y_2) = \tau(Y_2, Y_1)$, Lee suggested the symmetric measure $\tau = \mathrm{Max}[\tau(Y_1, Y_2), \tau(Y_2, Y_1)]$. This measure is one when $\tau(Y_1, Y_2)$ or $\tau(Y_2, Y_1)$ is one. However, a maximum correlation of one should only appear when both $\tau(Y_1, Y_2)$ and $\tau(Y_2, Y_1)$ showed a correlation of one. Therefore, it is more appropriate to define $\tau = [\tau(Y_1, Y_2) + \tau(Y_2, Y_1)]/2$.

It can be shown that nominal variables result in similar problems as with ordinally scaled variables. Their treatment is analogous to that presented in Sections 6.2 and 6.3. Unfortunately, the method from Section 6.4 cannot be applied. The reason is that a measure for nominal variables is invariant with respect to permutations of the categories, while this does not apply to the correlation coefficient generally.

Although a lot of care was spent on the simulation of tables with nominal pairwise association measures, it seems that the meaning of such scenarios is limited. In practice, when an observed table is analysed, it is more important to simulate either tables under a null hypothesis or tables under different alternative hypotheses. In both cases, the two-way marginal totals can be viewed as fixed. With given two-way marginal totals, the two-way associations can be determined. (When there are $c$ variables, even the $c$-way marginal totals can be viewed as fixed.) Then, it remains to define the properties of the table to simulate and to determine the cell properties. The simulation can then be carried out with the inversion method of Lee [35].

The statements resulting from such simulations are normally about the effect of certain properties of a table. This is only correct when there is just one table with the properties. Commonly, there are several such tables; i.e., it is necessary to simulate at least some extreme tables (where the cell probabilities are edges of the convex set of admissible tables) and an average table (e.g., the table with the given restraints and maximum entropy).

**Table 12. The $\chi^2$ values for certain interaction hypotheses concerning the data of Mood (1950).**

| Method | Null hypothesis | | |
|---|---|---|---|
| | Mutual independence | C independent of A and B | Zero three-way interaction |
| | $D_{A,B} = D_{A,C} = D_{B,C} = 0$ | $D_{A,C} = D_{B,C} = 0$ | $D = 0$ |
| Mood [42] | 110.1 | 86.7 | - |
| Lancaster [21] | 132.0 | 107.9 | 7.80 |
| Snedecor [44] | 132.0 | 93.7 | 19.57 |
| Cheng [43] | 120.6 | 96.4 | 6.82 |
| Maximal entropy | 132.0 | 93.7 | 6.80 |

Marked cells contain (nearly) correct results.

In this study, Good's [27] investigations on maximum entropy under restraints were repeatedly used. It allows us to determine hypothetical tables without knowing the MLEs of the log- linear model explicitly, as it suffices to formulate the equations of the hypotheses.

The 2×2×2 data of Mood [42] were repeatedly used to demonstrate improved theories. The observations were $n_{1,1,1} = 79$, $n_{1,1,2} = 73$, $n_{1,2,1} = 62$, $n_{1,2,2} = 168$, $n_{2,1,1} = 177$, $n_{2,1,2} = 81$, $n_{2,2,1} = 121$ and $n_{2,2,2} = 75$. Application of the maximum entropy principle led to the results in Table 12. Due to the concavity of entropy, convergence was excellent. Comparison with results of previous theories shows the appropriateness of the principle. A comment on the results [43] is given below.

Note that the row for maximum entropy satisfies the theoretical results of Roy and Kastenbaum [39] on the MLE for the log-linear model with given zero-, one-, and two-way margins. The approach [43] yielded the correct result only in one case.

Due to the concavity of entropy, the maximum is a global one. This explains why Bartlett's criterion, which is a cubic equation, has one real and two complex solutions under all admissible circumstances.

This fact makes it easy to prove that Bartlett's and Bennett's criteria agree if $p_1 = p_2 = p_3 = 1/2$ holds or if at least two of the three two-way interactions are zero. When these conditions are substituted into (17), the criterion (15) with the appropriate cell frequencies can be expanded. The result is in both cases

$$D = L_{1,2,3}. \tag{53}$$

Good's [27] paper on maximum entropy under restraints, however, was sometimes overlooked. For example, Streitberg [25, 26] did not include this approach in his contemplations. When Fienberg and Rinaldo [2] cited Good, they did not write about entropy, and when Fienberg and Rinaldo [30, 31] wrote about entropy, they did not cite Good. The authors of [43] wrote about entropy without recognizing Good's results. They stated that an equivalence test for the independence between one variable and the remaining two in [39] was not correct. For a 2×2×2 table, the statement of [39] can be formulated as follows: Assuming $D_{1,3} = D_{2,3} = 0$, the validity of Bartlett's criterion, i.e., $D = 0$, is equivalent to $p_{i,j,k} = p_{\bullet,\bullet,k} \, p_{i,j,\bullet}$ for $i,j,k \in \{1,2\}$.

Applying $D_{1,3} = D_{2,3} = 0$ to (16), we get via (53)

$$D = p_{1,1,1} - (p_1 \, p_2 \, p_3 + p_3 \, D_{1,2}) = p_{1,1,1} - p_3(p_1 \, p_2 + D_{1,2}) = p_{1,1,1} - p_3 \, p_{1,2}. \tag{54}$$

A proof for Roy and Kastenbaum's [39] statement is now given. Starting the proof with $D = 0$, we get $p_{1,1,1} = p_3 \, p_{1,2}$. Substituting $p_3 \, p_{1,2}$ for $p_{1,1,1}$ in (11), together with $p_{1,3} = p_1 p_3$ and

$p_{2,3} = p_2 p_3$, gives

$$\mathbf{p}(p_{1,1,1}) = \begin{bmatrix} p_{1,1,1} \\ p_{1,1,2} \\ p_{1,2,1} \\ p_{1,2,2} \\ p_{2,1,1} \\ p_{2,1,2} \\ p_{2,2,1} \\ p_{2,2,2} \end{bmatrix} = \begin{bmatrix} p_3\, p_{1,2} \\ (1-p_3)p_{1,2} \\ p_3(p_1 - p_{1,2}) \\ (1-p_3)(p_1 - p_{1,2}) \\ p_3(p_2 - p_{1,2}) \\ (1-p_3)(p_2 - p_{1,2}) \\ p_3(1 - p_1 - p_2 + p_{1,2}) \\ (1-p_3)(1 - p_1 - p_2 + p_{1,2}) \end{bmatrix} = \begin{bmatrix} p_{\bullet,\bullet,1}p_{1,1,\bullet} \\ p_{\bullet,\bullet,2}p_{1,1,\bullet} \\ p_{\bullet,\bullet,1}p_{1,2,\bullet} \\ p_{\bullet,\bullet,2}p_{1,2,\bullet} \\ p_{\bullet,\bullet,1}p_{2,1,\bullet} \\ p_{\bullet,\bullet,}2p_{2,1,\bullet} \\ p_{\bullet,\bullet,1}p_{2,2,\bullet} \\ p_{\bullet,\bullet,2}p_{2,2,\bullet} \end{bmatrix} ; \qquad (55)$$

i.e., indeed $p_{i,j,k} = p_{\bullet,\bullet,k}\, p_{i,j,\bullet}$ holds for $i,j,k \in \{1,2\}$.

Starting with $p_{1,1,1} = p_2\, p_{1,2}$ and regarding (54), we immediately get $D = 0$. Therefore, the statement in [43] was not correct. This caused the wrong results in Table 12.

Another advantage of the entropy principle is that there are no problems with zero counts. As Khinchin [29] noted, an event with probability zero need not be considered. It might be viewed as mythical to exclude an event from a contingency table, but this view is overcome when the table is considered as a multinomial distribution. In that case, the dimension simply reduces. In this way, there are also no problems with the evaluation of the $\chi^2$ – or $G^2$ – test statistics, since singularities cannot appear.

Hence, a numerical procedure that maximizes entropy should test whether a probability $p_i$ is larger than, say, $10^{-8}$. Otherwise, the term $p_i \ln p_i$ is set to zero when summing up entropy via (2).

The concept of entropy has a great meaning in thermodynamics. There, a system drives to an equilibrium state, one with maximum entropy. Similar processes are observable in population genetics, where large populations with random mating converge to independence of genotypes, even for closely linked loci. (Only the one-way margins are maintained.) The obtained state is named the linkage equilibrium, while the presence of a two-way interaction is called the linkage disequilibrium. In population genetics, there are also events that decrease entropy, such as mutations, inbreeding, and selection. While the aspects of processes concerning populations are complex, they are simple compared with social or metabolic processes. In human society, there are forces toward increase of entropy and forces toward reduction of entropy, from the smallest groups up to the human race.

Hence, analyzing disequilibria using contingency tables encompasses the task of thinking about forces that affect a process.

In this study, two points were repeatedly applied: the maximization of entropy and the treatment of a contingency table as a multinomial distribution. The question arises whether entropy has an analytical relation to the likelihood function of a multinomial distribution.

The $k$-dimensional multinomial probability distribution is $L = n!/(\prod_{i=1}^{k} n_i!) \prod_{i=1}^{k} p_i^{n_i}$. In the ideal case $p_i = n_i/n$ (which is at least asymptotically satisfied), the log-likelihood of the factor $\prod_{i=1}^{k} p_i^{n_i}$ is $n \sum_{i=1}^{k} p_i \ln p_i = -nH$. This relation suggests that the maximal likelihood is related to the minimal entropy and the maximum entropy to the minimal likelihood. In the given context, however, the observations $n_i$ underlie constraints, such as the given one- and two-way marginal totals. Therefore, the multinomial coefficient $n!/(\prod_{i=1}^{k} n_i!)$ is no longer a

constant. The application of Stirling's formula $\ln n! \sim n \ln n - n + \ln\sqrt{2\pi n}$ leads to

$$\ln \frac{n!}{\prod_{i=1}^{k} n_i!} \sim \ln \sqrt{\frac{n}{(2\pi)^{k-1} \prod_{i=1}^{k} n_i}} + n\,H. \tag{56}$$

Therefore,

$$\ln L \sim \ln \sqrt{\frac{n}{(2\pi)^{k-1} \prod_{i=1}^{k} n_i}} \tag{57}$$

is an asymptotic expansion of the log-likelihood function. The likelihood function is then

$$L \sim \sqrt{\frac{n}{(2\pi)^{k-1} \prod_{i=1}^{k} n_i}} = \sqrt{\frac{1}{(2\pi n)^{k-1} \prod_{i=1}^{k} p_i}}, \tag{58}$$

where the right-hand-side expression corresponds to the formulas given in [45].

Leaving out the constants, the derivative of the likelihood function is

$$\frac{d}{dx} L = \frac{d}{dx} \sqrt{\frac{1}{\prod_{i=1}^{k} n_i}} = \frac{d}{dx} \prod_{i=1}^{k} n_i^{-1/2} = \sum_{i=1}^{k} \left(-\frac{1}{2}\right) n_i^{-3/2} \frac{dn_i}{dx} \prod_{j \neq i} n_j^{-1/2} = -\frac{1}{2} \frac{\sum_{i=1}^{k} \frac{1}{n_i} \frac{dn_i}{dx}}{\sqrt{\prod_{i=1}^{k} n_i}}. \tag{59}$$

The constraints investigated in this study, given one-, two-, or three-way marginal totals, lead to cells $n_i$ or $p_i$, which are linear combinations of free parameters and given constants. Actually, each free parameter $x$ appears in the cells either as $x$ or $-x$; see Eqs (4), (11), (17) and Table 2. Therefore, $dn_i/dx$ is either 1 or -1. Thus, the derivative (59) is zero if

$$\sum_{+} \frac{1}{n_i} = \sum_{-} \frac{1}{n_i} \tag{60}$$

is satisfied. The plus sign means that summation has to be taken over all cells where $x$ appears as $+x$, and the minus sign means that summation has to be taken over all cells where $x$ appears as $-x$. Both cases appear with the same frequency. Therefore, Eq (60) indicates that the agreeing harmonic means guarantee an optimum. It can be shown that the second derivative is strictly positive; i.e., solving Eq (60) gives the value $x$ for which the likelihood is minimal.

Two examples of 2×2 tables are presented in Fig 1.

One can see in Fig 1 that the minimum likelihood corresponds to the maximum entropy. For the first example, the condition (60) for the minimum of $L$, and therefore also for $\ln L$, is

$$\frac{1}{x} + \frac{1}{x-9} = \frac{1}{21-x} + \frac{1}{16-x}. \tag{61}$$

The solution is $x \approx 12.3$, while the maximum entropy appears for $x = 12$. The second example particularly shows the goodness of the asymptotic expression, as it nearly agrees with the exact one.

The asymptotic condition for the minimum likelihood (60) has a special relation to the maximum entropy principle. When Good [27] determined the maximum entropy in the same context as here, he found the condition

$$\prod_{+} n_i = \prod_{-} n_i. \tag{62}$$

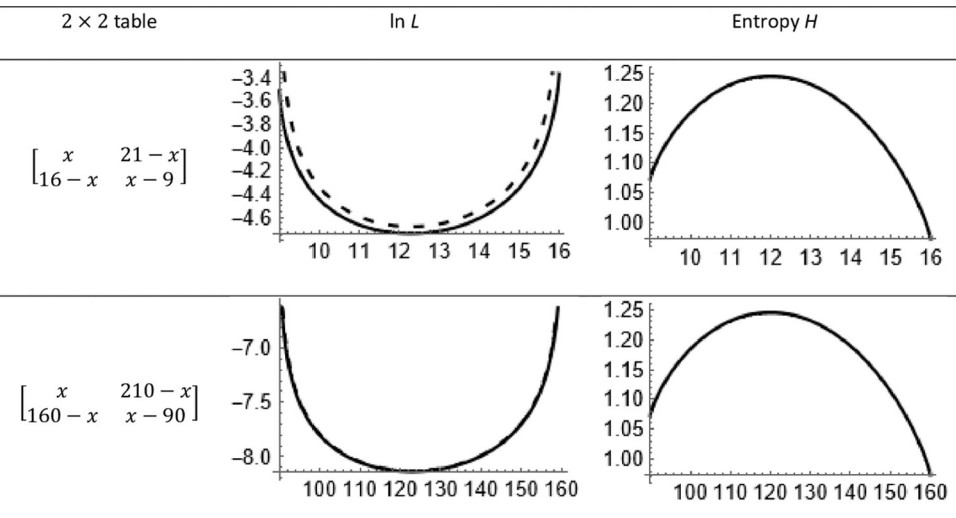

**Fig 1. The log-likelihood function ln L and the entropy H for the free parameter x of two 2×2 tables.** The broken lines correspond to the asymptotic expression (57).

A comparison of this condition with condition (60) proves that the equality of geometric means applies for maximizing the entropy while the equality of harmonic means applies for minimizing the likelihood.

The most elementary restraint is that the number of observations is just $n$. Then, we can write $n_k = n - \sum_{i=1}^{k-1} n_i$, and both Eqs (60) and (62) give the same results $n_k = n_i$, $i = 1,2,\cdots,k-1$. From this, $n_i = n/k$ and $p_i = 1/k$ result for $i = 1,2,\cdots,k$; i.e. the maximum entropy solution is identical with the asymptotic minimum likelihood solution.

## 9 Conclusions

Five methods contributed to a considerable improvement in the theory of contingency tables: (1) the use of the LD measure, (2) the treatment of a table as a multinomial distribution, (3) the use of algebraic software, (4) the consequent utilization of linear programming, and (5) the application of the maximization of entropy under restraints.

Using the linkage disequilibrium parameter $D$ as a measure of association between two categorical variables, which is essentially the determinant $D = p_{11}p_{22} - p_{12}p_{21}$ of a 2×2 table, sufficed to quantify Simpson's paradox. The difference between a two-way interaction and the averaged partial interactions for the categories of a third variable was derived. For a 2×2×2 table, the difference was $D_{1,2} - \bar{D}_{1,2|3} = D_{1,3}D_{2,3}/[p_3(1 - p_3)]$. It became particularly clear that the agreement of $D_{1,2}$ and $\bar{D}_{1,2|3}$ can only arise when the third variable is independent of the first or second one (because $D_{i,j} = 0$ means independence between variables $i$ and $j$).

In many experiments, there is one response variable together with two explanatory variables. The latter ones can be arranged to ensure $D_{2,3} = 0$, for example, by applying the treatments to the same fraction of males and females. In this way, the difference $D_{1,2} - \bar{D}_{1,2|3}$ is zero and Simpson's paradox is circumvented.

However, with unplanned experiments or with two or three response or random variables, Simpson's paradox, which is essentially $D_{1,2} \neq \bar{D}_{1,2|3}$, is to be expected.

It is of particular interest that, with knowledge of merely one- and two-way parameters, implications for the three-way structure are possible. This was demonstrated with the Berkeley data and could be shown with other real data reflecting Simpson's paradox.

With the log-linear model, a variety of important hypotheses can be tested. However, practically relevant hypotheses were not the focus. One such hypothesis is that the actual degree of interaction between two categorical variables is the same within all levels of another categorical variable. This study derived a model for this hypothesis and applied it to the Berkeley data. It was possible to show that the model of agreeing associations between sex and admittance (measured with Pearson's correlation coefficient) within the departments does not fit the data. However, with a refinement, it could also be shown that just one department caused the heterogeneity. Within the other departments, there was no significant association between sex and admittance.

There is a need for further investigations into an optimal systematic strategy to find a parsimonious model. However, the new approach presented here is suitable for answering important questions and surely enriches the theory of contingency tables.

Tables with zero counts provoke the question of whether these counts appeared by chance or whether they were a necessary consequence of the given two- or more-way marginal totals. The application of linear programming provided a much simpler and successful way to obtain fixed zero counts than other methods used so far. Furthermore, fixed nonzero counts can be determined; thus, the number of independent variables (degrees of freedom) could be further reduced.

Improvements were achieved for the simulation of categorical variables with given relationships, and the restrictions of formerly used procedures could be circumvented.

In this study, Good's [27] investigations on maximum entropy under restraints were repeatedly used. Maximizing entropy under restraints means determining the table or multinomial distribution that is characterized by a minimum of information, the largest disorder, or as-uniform-as-possible cell frequencies under given assumptions. It allows the numerical determination of hypothetical tables by incorporating the equations of the hypotheses. It was recalled that, with appropriate hypotheses, the results of the maximum entropy principle agree with those of the MLEs of the log-linear model. Recent doubts about the validity of hierarchical log-linear models could be eliminated.

The relation between Bartlett's multiplicative and Bennett's additive measure of the three-way interaction was investigated. As it turned out, Bennett's measure is a simplified version of the first-order Taylor expansion of Bartlett's measure. Since Bartlett's measure (which is in concordance with the maximum entropy principle and with the log-linear model) has a deeper meaning than Bennett's measure, it is concluded that Bartlett's measure is the first choice. When an easy-to-calculate measure is preferred, the full first-order Taylor expansion should be applied instead of Bennett's measure.

It was shown for contingency tables that the concept of entropy is related to the likelihood principle for the multinomial distribution. In particular, a hypothetical table with maximum entropy under linear restraints (like the given marginal totals) and a table with minimum likelihood under the same restraints are similar but not identical. The tables at the bounds of the admissible region yield local minima of entropy and local maxima of the likelihood function.

It is hoped that applicants feel encouraged to test not only the classical hypotheses but also those of particular interest and that theoreticians further improve the suggested methods.

## Acknowledgments

The author thanks two referees of an earlier version of this manuscript for valuable suggestions.

## Author Contributions

**Investigation:** Friedrich Teuscher.

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
