## [Decision Letter · Decision Letter 0]

27 Oct 2021

PONE-D-21-12542

The quantification of Simpson’s paradox and other contributions to contingency table theory

PLOS ONE

Dear Dr. Teuscher,

Thank you for submitting your manuscript to PLOS ONE. After careful consideration, we feel that it has merit but does not fully meet PLOS ONE’s publication criteria as it currently stands. Therefore, we invite you to submit a revised version of the manuscript that addresses the points raised during the review process.

We look forward to receiving your revised manuscript.

Kind regards,

Hanna Landenmark

Senior Editor, PLOS ONE

on behalf of 

Wei Zhu

Journal Requirements:

Additional Editor Comments (if provided):

Dear Author, we have reviewed your paper and found it well written and worthy to be published at PLOS ONE.

Reviewers' comments:

Reviewer's Responses to Questions

**Comments to the Author**

1. Is the manuscript technically sound, and do the data support the conclusions?

Reviewer #1: Yes

2. Has the statistical analysis been performed appropriately and rigorously? 

Reviewer #1: Yes

3. Have the authors made all data underlying the findings in their manuscript fully available?

Reviewer #1: Yes

4. Is the manuscript presented in an intelligible fashion and written in standard English?

Reviewer #1: No

5. Review Comments to the Author

Reviewer #1: The study “The quantification of Simpson’s paradox and other contributions to contingency table theory” is interesting.

In this study, the authors claimed that, to date, there is no test available that determines whether the partial interactions of a certain variable agree or disagree, and the presented work closes this gap. This work reveals the relation of the multiplicative and the additive measure of a three-way interaction. Another contribution addresses the question of which cells in a contingency table are fixed when the first- and second-order marginal totals are given. The proposed procedure not only detects fixed zero counts but also fixed positive counts. This impacts the determination of the degrees of freedom. Furthermore, limitations of methods that simulate contingency tables with given pairwise associations are addressed.

The paper is well set, and the contents are clearly described. The authors almost achieved their objectives. However, the following suggestions should be incorporated before resubmitting the paper.

1. Lines 493, correct the equation 41. Some entries are doubled/missing.

2. Special attention must be given to the language as well.

3. A new section of the conclusion must be added to the paper which concludes the whole study.

4. To broaden the scope of the study the authors are needed to add some more recent references.

6. PLOS authors have the option to publish the peer review history of their article (what does this mean?). If published, this will include your full peer review and any attached files.

Reviewer #1: No

---

## [Author Response · Author response to Decision Letter 0]

24 Dec 2021

Dear editor, dear reviewer,

I sent you the revision of my manuscript “The quantification of Simpson’s paradox and other contributions to contingency table theory”.

All changes are marked in the manuscript.

Points raised by the reviewer:

1 Lines 493, correct the equation 41. Some entries are doubled/missing. This was caused by shifted line numbers. I found a solution for that.

2 Special attention must be given to the language as well. I paid $1667 for professional edition of the last version and $266 for this revision (for the new part of the discussion section and the conclusion section). Since in between I made not much changes, there should not be very much mistakes. Please let me know, what should be corrected.

3 A new section of the conclusion must be added to the paper which concludes the whole study. I added a conclusion section.

4 To broaden the scope of the study the authors are needed to add some more recent references. I added two more recent references. 

Meanwhile I found it worthwhile to investigate the relation of the maximum entropy principle and the likelihood principle applied to contingency tables with given restraints. The results are interesting and I added them at the end of the discussion section.

(I did not find literature about it. Only two misleading texts in the www. If I overlooked something, let me know it please.

There is literature on entropy and likelihood within the Bayesian framework. Nevertheless, this is far away from the topics of the study. So I do not cite it.)

With the writing of the conclusion section, I shifted some parts of the discussion section (and a few others).

The name of my institute changed. (I am retired but got the opportunity to publish, to use computers, email etc.)

Further changes are mainly the edition of sub-headlines.

I think the manuscript is better now. I hope you see it similar.

Sincerely yours, Friedrich Teuscher

Sincerely yours, Friedrich Teuscher

---

## [Editor Report · Decision Letter 1]

27 Dec 2021

The quantification of Simpson’s paradox and other contributions to contingency table theory

PONE-D-21-12542R1

Dear Dr. Teuscher,

We’re pleased to inform you that your manuscript has been judged scientifically suitable for publication and will be formally accepted for publication once it meets all outstanding technical requirements.

Kind regards,

Wei Zhu, Ph.D.

Academic Editor

PLOS ONE
---

## [Editor Report · Acceptance letter]

8 Feb 2022

PONE-D-21-12542R1 

The quantification of Simpson’s paradox and other contributions to contingency table theory 

Dear Dr. Teuscher:

I'm pleased to inform you that your manuscript has been deemed suitable for publication in PLOS ONE. Congratulations! Your manuscript is now with our production department. 

Kind regards, 

on behalf of

Dr. Wei Zhu 

Academic Editor

PLOS ONE